

# Probing black hole microstate evolution with networks and random walks

**Anthony M. Charles[1,2][⋆] and Daniel R. Mayerson[1,3][†]**

**1** Department of Physics and Leinweber Center for Theoretical Physics,
University of Michigan, 450 Church Street, Ann Arbor, MI 48109-1020, USA
**2** Institute for Theoretical Physics, KU Leuven,
Celestijnenlaan 200D, B-3001 Leuven, Belgium
**3** Institut de Physics Théorique, Université Paris Saclay
CEA, CNRS, F-91191 Gif-sur-Yvette, France

⋆ anthony.charles@kuleuven.be, † daniel.mayerson@ipht.fr

## Abstract

We model black hole microstates and quantum tunneling transitions between them with networks and simulate their time evolution using well-established tools in network theory. In particular, we consider two models based on Bena-Warner three-charge multi-centered microstates and one model based on the D1-D5 system; we use network theory methods to determine how many centers (or D1-D5 string strands) we expect to see in a typical late-time state. We find three distinct possible phases in parameter space for the late-time behaviour of these networks, which we call ergodic, trapped, and amplified, depending on the relative importance and connectedness of microstates. We analyze in detail how these different phases of late-time behavior are related to the underlying physics of the black hole microstates. Our results indicate that the expected properties of microstates at late times cannot always be determined simply by entropic arguments; typicality is instead a highly non-trivial, emergent property of the full Hilbert space of microstates.



# 1  Introduction and Summary

String theory is expected to somehow resolve puzzles arising in general relativity involving black holes, such as the origin of their entropy and the information paradox. The fuzzball programme argues that the resolution of such puzzles is that extended stringy objects alter the horizon structure of black holes drastically from its classical expectation. In this picture, the black hole horizon should be seen as an effective geometry, averaged over the individual "fuzzballs" that actually make up the states of the black hole [1–7].

Microstate geometries are fuzzballs that can be constructed and studied as classical solutions in the supergravity limit of string theory; these are smooth, horizonless solutions with the same asymptotic charges as the black holes. These microstate geometries intrinsically live in dimensions larger than four and have non-trivial topological cycles that are supported by fluxes. This non-trivial topological structure allows for smooth supersymmetric soliton solutions that support charges and mass [8, 9]. For the three-charge supersymmetric black hole, many microstate geometries have already been constructed in the literature. These include

the multi-centered Bena-Warner solutions [2] and the more recent superstrata [10–12], which are themselves a generalization of the two-charge D1-D5 Lunin-Mathur supertubes [1, 13].

Assuming supersymmetry simplifies the search for microstate geometries, as the relevant BPS equations are often more tractable than the equations of motion themselves.[1] Importantly, these supersymmetric states are void of actual dynamics and so they necessarily avoid the question: *how can smooth black hole microstates form in a dynamical process?* A gravitational collapse of a shell of matter will form a horizon well before any curvatures or quantum effects are expected to be large, which seems to be in contradiction to the fuzzball proposal where no horizon should form.

A resolution to this puzzle, proposed in [24–26], is that the *large phase space* at horizon scales is the crucial ingredient that invalidates the usual classical intuition and renders quantum effects large at the horizon scale. The logic is that a collapsing shell of matter can quantum tunnel into a fuzzball microstate before it forms a horizon. The tunneling amplitude to form any one particular fuzzball is $\mathcal{O}(e^{-S})$, exponentially suppressed by the would-be black hole entropy. However, when the collapsing shell reaches its putative horizon scale, the available number of microstates is incredibly large, namely $e^S$. The two exponentials can cancel, and the tunneling probability to go to *any* microstate ends up being $\mathcal{O}(1)$. The result is that the collapsing shell of matter will necessarily quantum tunnel into a fuzzball instead of forming a horizon.

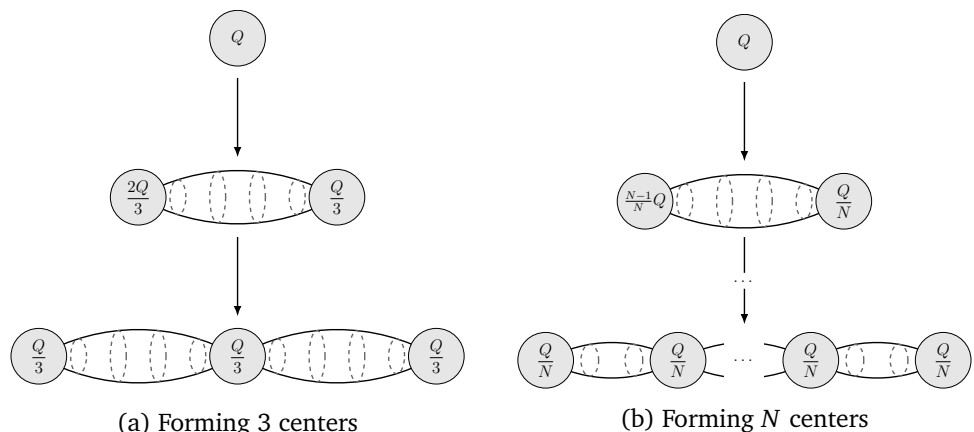

(a) Forming 3 centers

(b) Forming $N$ centers

Figure 1: Two sample formation paths to tunnel into a multi-centered microstate.

These arguments are very general, but they are not based on any explicit calculations involving actual fuzzballs or microstate geometries. The first (and so far, only) concrete calculation of a formation rate of black hole microstates by quantum tunneling was performed in [27]. There, the authors considered forming smooth multi-center microstate geometries by starting with a collapsing shell of branes and repeatedly tunneling off a small amount of charge from the shell onto a new center. This iterative tunneling procedure is depicted schematically in figure 1. By treating the tunneling branes in this picture as probe branes, the tunneling amplitudes can be computed explicitly from the probe brane action. The final result is that the tunneling amplitude to end up in a final state with $N$ centers goes like

$$\Gamma \sim \exp\left(-\alpha_0 S_{BH} N^{-\beta}\right), \qquad (1)$$

where $\alpha_0$ is a microstate-dependent number, the black hole entropy is $S_{BH} = 2\pi Q^{3/2}$ (for three

---

[1]See [14–21] for explicit constructions of non-supersymmetric microstate geometries. It is also possible to construct non-supersymmetric microstates by adding a non-supersymmetric probe onto a supersymmetric background [22, 23].

equal electric charges), and where $\beta$ is a positive exponent of order one.[2] The conclusion is that the tunneling rate is enhanced for larger values of $N$, and so it is easier to form a microstate geometry with a large number of centers.

Of course, this only tells us what to expect when the collapsing shell of matter first tunnels into a microstate. After this initial tunneling, it is still possible for tunneling between various microstates to occur as time goes on. A broader question we can ask is the following: *what do we expect to see after the collapsing shell of matter "settles down"?* That is, if we wait for a long enough time for the system to end up in some kind of equilibrium configuration, what microstate should we expect the system to be in? The calculations of [27] are not enough to answer this question, because they only consider particular formation paths. It may be relatively easy to form a microstate with a large number of centers along these paths, but if there are many more few-centered microstates available (with formation paths leading to them) in the phase space, then the complete picture may result in being more likely to end up with a small number of centers after all. To have such a complete picture of the late-time dynamics of the microstates, we should take into account all possible microstates (including possible fuzzballs that do not have a classical microstate geometry description, or that do not have one in the duality frame the evolution started out in [28, 29]), their degeneracies, and their interactions (in the form of quantum tunneling between each other).

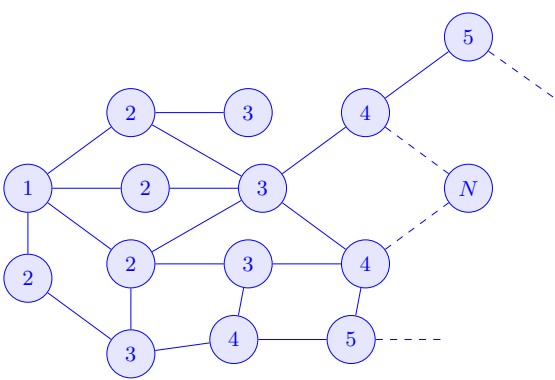

Figure 2: A schematic network of microstates, labelled by their numbers of centers, with many different tunneling transitions between states possible.

This very naturally leads us to consider a *network*, as depicted in figure 2, where every node in the network corresponds to a particular microstate, every edge corresponds to an allowed tunneling path, and these edges are weighted by the corresponding tunneling rate between the nodes. We can then use the full array of well-developed tools in network theory [30] to understand the late-time behavior of such a network. For example, the late-time relative importance of nodes within a network is directly related to the *eigenvector centrality* of the network, as we will discuss in section 2.

It is important to note that we cannot feasibly construct all possible microstate geometries (let alone non-classical fuzzballs) explicitly, let alone compute the transition rates between each pair. Therefore, the networks we construct in this paper will necessarily be models, in the sense that we will try to capture important general features of the microstates, while leaving out some of the more intricate details of the actual microstate geometries. Our models will have a number of a priori unknown parameters that will parametrize how the degeneracies of the microstates and the tunneling rates between them depend on certain properties of the microstates. By exploring the phase space of these parameters, we will be able to make general statements about which microstate properties are important in determining the late-time behaviour of the black hole microstate evolution.

---

[2]Specifically, $\beta = 3/2$ and $\beta = 0.93$ for the two types of microstates considered in [27].

The rest of the paper is organized as follows. In the next subsection 1.1, we introduce the three network models we will consider and discuss how they capture features of certain classes of black hole microstates in string theory. In subsection 1.2, we briefly summarize our main results. In section 2, we provide an overview of how network theory methods and random walks can be used to understand the dynamics of quantum systems. In sections 3, 4, and 5, we present a detailed description of each of our three network models (respectively), use network theory methods to understand their time evolution, and then give a detailed discussion of the results. Finally, we conclude in section 6 with some important discussions regarding all three models.

We give more technical details on the network and random walk methods used throughout this paper in appendix A, and in appendix B we present a number of explicit calculations involving black hole microstates which were used to inform various features of our network models.

## 1.1 From Microstates to Network Models

Here, we introduce the three models of microstate networks we will study in this paper, and we discuss how they are inspired by existing classes of black hole microstate solutions in string theory.

### 1.1.1 Models 1 and 2: Multi-centered black hole microstates

Our goal for the first two models is to model the dynamics of five-dimensional three-charge, smooth, multi-centered microstate geometries [2], as reviewed in appendix B. These can be viewed as microstates of the five-dimensional three-charge supersymmetric BMPV black hole. Since we want non-trivial dynamics, we want to consider adding a small amount of non-extremality to these microstates in order to excite them away from BPS limit. The slightly non-extremal microstates obtained in this way should then be interpreted as microstates of a near-extremal BMPV black hole. Note that the amount of non-extremality we add to the microstate is not a tuneable parameter; we consider it to be infinitesimal to avoid large backreaction on the BPS microstates.

It is not known how many ways there are to add a small amount of non-extremality to a generic multi-centered black hole microstate geometry.[3] One possibility we can imagine is to give one or more of the centers small velocities relative to the others. Another possibility is to "wiggle" the bubbles themselves and excite oscillation modes of the topological bubbles between the centers. From these considerations, it seems very likely that different supersymmetric microstates have a different number of ways to excite them above extremality. We will allow for this possibility by taking into account a degeneracy factor for each microstate that depends on its properties.

The only dynamics that we allow in our models are the quantum tunneling transitions between different microstate geometries. Heuristically, the transitions we want to allow should be thought of as taking some or all charge from a center and tunneling it either onto a new center or an existing center, similar in spirit to the explicit tunneling calculations performed in [27]. This means any single transition will either leave the number of centers unaltered or change the number by one.

The Bena-Warner multi-center microstate geometries are complicated solutions in five-dimensional supergravity. The charges of the centers and their positions must satisfy the non-linear *bubble equations*, which become increasingly complicated to solve when the number of centers increases. Restrictions are often placed on the centers to facilitate solving the bubble

---

[3]One can add probes to supersymmetric backgrounds that will break supersymmetry [22, 23, 31, 32], but even a counting of all such possible non-extremal probes has not been done.

equations, such as putting them all on a single line. For our networks, we will use simple models that capture some of the important qualitative physics of these microstate geometries without having to actually construct all relevant supergravity solutions.[4]

Our first model is a very minimal model of black hole microstates where we only consider the number of centers $N$ that each microstate has. We will not assume any particular configurations of these centers, such as restricting them to be on a line. The degeneracy of each microstate (i.e. the number of ways to lift the microstates off extremality, as discussed above) can then only depend on $N$, while the tunneling rate between states can depend on the number of centers of the initial and final states.

Our second model has more features to allow for richer physics. To construct a microstate, we fix a total charge $Q$ of the black hole and divide this over a variable number $N$ of centers. For simplicity, we will only consider one type of charge in the system. We will further assume that the centers are all on a line. A microstate in model 2 is thus determined by giving an ordered list $\{Q_1, Q_2, \ldots Q_N\}$ of charges of each of the centers such that $\sum_{i=1}^{N} Q_i = Q$; this is called a *composition* of the integer $Q$. The degeneracy and transition rates are more intricate in model 2, as they can depend on the details of the charge distribution within the microstates. Note that the microstates in this model are in one-to-one correspondence with compositions of $Q$, and there are $2^{Q-1}$ such compositions.

### 1.1.2 Model 3: the D1-D5 system

In model 3, we want to understand dynamics of microstates in the D1-D5 system. Specifically, we will consider type IIB string theory on $\mathbb{R}^{4,1} \times S^1 \times T^4$ with $N_1$ D1-branes wrapping the $S^1$ and $N_5$ D5-branes wrapping $S^1 \times T^4$. This system is well-understood in many contexts.[5] In particular, the low-energy world-volume dynamics of this system are described by a sigma model CFT whose target space is a deformation of the symmetric product $(T^4)^N/S_N$, where $N = N_1 N_5$. The Ramond ground states of this CFT are holographically dual to the smooth Lunin-Mathur supertubes [1, 13].

We will find the "string gas" picture of the D1-D5 ground states the most useful, where these ground states can be seen as a gas of strings winding the $S^1$ with total winding number $N$. If there are $N_w$ strings with winding number $w$, then we must have

$$\sum_{w=1}^{\infty} w N_w = N. \tag{2}$$

Furthermore, there are 8 bosonic and 8 fermionic modes that each string can be in:

$$N_w = \sum_{\mu} N_{w,\mu} + \sum_{\mu'} N'_{w,\mu'}, \tag{3}$$

where the sum over $\mu$ is over the 8 bosonic modes (so $N_{w,\mu} = 0, 1, 2, \ldots$) and the sum over $\mu'$ is over the 8 fermionic modes (so $N'_{w,\mu'} = 0$ or 1). For a given $N_w$, there are $\Omega(N_w)$ distinct

---

[4]Given how complicated the Bena-Warner multi-centered geometries are, one might wonder if it is actually possible to find solutions that are related by the "splitting" transitions discussed above. We give a proof of concept of this in appendix B.2 by explicitly constructing two solutions to the bubble equations that have the same asymptotic charges and only differ by the centers that have undergone this splitting.

[5]For relevant overviews and discussions of the D1-D5 CFT, see e.g. [1, 33, 34].

ways of dividing the strings into the 8 bosonic and 8 fermionic modes, where:[6]

$$\Omega(k) = \sum_{l=0}^{8} \binom{8}{l} \binom{k-l+7}{7}. \tag{4}$$

In model 3, we will consider ground states in the D1-D5 system with a slight amount of non-extremality added. Using the string gas picture described above, we will model these states by unordered collections $\{w_1, w_2, \ldots\}$ of winding numbers, with $\sum_i w_i = N$. We will associate the correct winding mode degeneracies (4) to each microstate, but unlike in models 1 and 2 we will not associate an additional degeneracy related to the number of ways to add the slight non-extremality to the microstate. Note that an unordered collection $\{w_1, w_2, \ldots\}$ corresponds to a *partition* of the integer $N$; asymptotically, the number of partitions $p(N)$ scales as $p(N) \sim \exp\left(\pi\sqrt{2N/3}\right)$. Of course, the total number states we are considering in model 3, including the degeneracies (4), is precisely (by construction) the number of D1/D5 ground states, which scales as $\sim \exp\left(2\pi\sqrt{2N}\right)$.

We will allow transitions where a single string of winding $w$ can split into two smaller strings with windings $w_a, w_b$ (with $w_a + w_b = w$), or the reverse where two strings with windings $w_a, w_b$ combine into a larger string of winding $w = w_a + w_b$. The transition rate for either of these processes will then depend non-trivially on the initial and final windings $w_a, w_b$ and $w = w_a + w_b$.

## 1.2 Summary of Results

Table 1: Summary of our main results for the late-time behavior of each microstate model.

| | Late-Time Behavior | | | Phase Transition? |
|---|---|---|---|---|
| | Ergodic | Trapped | Amplified | |
| **Model 1** (N centers) | × | | | No |
| **Model 2** (N centers with charges) | × | × | | Yes |
| **Model 3** (winding strings in D1-D5) | | | × | No |

| | Ergodic | Trapped | Amplified |
|---|---|---|---|
| Degeneracy important? | Yes | No | Yes |
| Transition rates important? | No | Yes | Yes |

In section 1.1, we have introduced three different network models of black hole microstates. (They will be further specified in sections 3.1, 4.1, and 5.1, respectively.) Our main goal in this work is to use network theory tools to study the time-evolution of these models. Crucially, the methods we will use do not assume the validity of equilibrium statistical mechanics, which allows for non-trivial and interesting late-time behavior. Our main results are shown in table

---

[6]To understand this formula, note that $l$ denotes the number of fermionic excitations turned on (which is limited to 8). The first factor is the combinatorial factor associated with distributing the $l$ fermionic excitations into 8 possible bins. The second factor is the combinatorial factor for dividing the $k-l$ bosonic excitations into 8 possible bins, with the possibility of putting multiple excitations into the same bin; i.e. the number of *weak compositions* of $k-l$ into 8 parts.

1. We find that there are, broadly speaking, three different types of late-time behavior that our microstate networks exhibit:

- **Ergodic behavior.** All microstates are approximately equally likely at late times, and so the probability distribution of microstates is determined entirely by the microstate degeneracies and not the details of tunneling rates between states. Random walks on the networks in this regime are able to move around the entire network freely.

- **Trapped behavior.** At late times, the probability distribution is restricted to only a small subnetwork of the full microstate network. The transition rates between microstates determine precisely what subnetwork is relevant. Random walks are effectively restricted to move only on this subnetwork.

- **Amplified behavior.** The most degenerate microstates comprise the most highly-connected nodes on the network. At late times, the system is much more likely to be on these highly-connected nodes than any others. Random walks are likely to stay on this highly-connected subnetwork, but excursions to other states are allowed.

Model 1 shows only ergodic behavior, while model 3 shows only amplified behavior. Model 2 can demonstrate either ergodic or trapped behavior, depending on where in parameter space we are. Interestingly, the cross-over between these two types of behaviors is very sharp and sudden, indicating a phase transition in parameter space of the network's late-time behavior.

As emphasized earlier, the main question we want to answer is *what do we expect to see at late times as these black hole microstate systems evolve*? If the system exhibits ergodic late-time behavior, this means that all microstates are (approximately) equally likely and so we can determine properties of a typical state simply by counting the number of states with that property. For example, in model 1 we can look at the number of microstates with a given number of centers $N$, and whichever value of $N$ maximizes this degeneracy will be the number of centers we expect a typical late-time state to have. If the system exhibits amplified late-time behavior, then these most degenerate (with respect to the number of centers or number of winding strings) microstates are also the most connected in terms of tunneling paths, and so this the system will favor these highly-degenerate states even more strongly. In the trapped phase, though, we cannot simply tabulate all microstates to understand typicality; the system at late times is *forced* to be in one of only a small subset of black hole microstates. This behavior is surprising because it indicates that entropic arguments are not sufficient to explain the dynamics of the system, despite its large ground-state degeneracy of microstates. We will elaborate on the impliciations of this trapped late-time behavior for Bena-Warner microstates in section 6.

The main takeaway from all of this is that the late-time dynamics of black hole microstates have a rich and interesting structure to them. We find that there is no simple way to express what "typicality" in black hole microstates means; it depends intricately on the full details of the Hilbert space of microstates. Moreover, the network theory methods presented in this work give an effective way to probe microstate dynamics, and we find a number of intriguing results that are worthy of further exploration.

## 2 Network Theory and Quantum Tunneling

Consider a quantum mechanical system with a discrete number of accessible states. Quantum fluctuations will generically give rise to non-zero tunneling probabilities between different states. The dynamics of the system are described by its Hamiltonian, which can be used to compute how a particular initial state (or probabilistic superposition of states) tunnels into

other states over time. However, these kinds of computations are difficult to perform in many cases, in particular for systems with a very large number of states. In addition, these methods fail if we only know estimates of the tunneling rates between states and not the full Hamiltonian.

A historically successful approach that evades some of these difficulties is to treat the time evolution of the system as a stochastic process, where time is discretized, and at each discrete time slice the system evolves randomly to another state according to the tunneling amplitudes that are available to the current state [35–37]. The dynamics of this stochastic system are then understood very naturally through the lens of network theory. Specifically, we can view these systems as directed networks where the nodes are the available states in the system and the edges between nodes are weighted by the corresponding tunneling amplitudes. Many properties of the underlying quantum system can then be understood quantitatively by analyzing properties of this corresponding network. In particular, this network-theoretic approach has led to many significant developments in physics-related fields, including percolation [38], protein interactions [39], quantum cosmology [40, 41], tunneling in the string landscape [42], and brain function [43], to name a few.

In this paper, we will be interested in asking the following question: *what state do we generically expect the system to be in*? That is, as the quantum system tunnels and evolves over time to some kind of equilibrium configuration, how can we compute the probability that the system is in any particular one of its many accessible states? These are hard problems to tackle for generic quantum systems, due to both the difficulty of numerically evolving the Schrödinger equation as well as the sensitivity of this evolution to initial conditions [44, 45]. However, network theory gives us a whole slew of powerful tools designed to answer exactly these sorts of questions. In particular, we will investigate *eigenvector centrality* and *random walks* of networks as tools to probe the late-time dynamics of quantum systems.

## 2.1 Eigenvector Centrality

Concretely, let's consider a system with $N$ distinct accessible states labelled by $i = 1, \ldots, N$, each with an associated degeneracy $\omega(i)$; for example, we could consider a system with $N$ distinct energy levels and $\omega(i)$ possible ways for the system to be in each energy level $i$. We denote the tunneling rate from state $i$ to state $j$ as $\Gamma(i \rightarrow j)$. Note that we will allow for self-transitions $\Gamma(i \rightarrow i)$ as well. This system can be represented by a network, as shown in figure 3, where the nodes of the network are the states and the edges are the tunneling amplitudes.

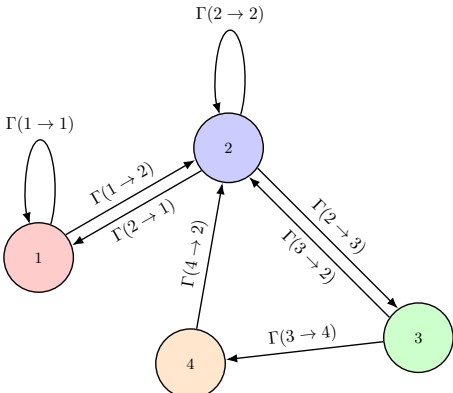

Figure 3: A sample of how a stochastic system can be represented with a network. Nodes correspond to states and directed edges correspond to allowed transitions; the edges are correspondingly weighted by the transition rate.

The adjacency matrix $\mathbf{A}$ of this network is defined as the $N \times N$ matrix whose elements $A_{ij}$ are the edge weights of the graph (i.e. the transition rates $\Gamma(i \to j)$), weighted by the degeneracy of the starting and ending nodes (i.e. the degeneracy of the initial and final states). That is,

$$A_{ij} = \omega(i)\Gamma(i \to j)\omega(j). \tag{5}$$

The degree $d_i$ of a node $i$ is the sum of all outgoing adjacency matrix elements from that node:

$$d_i = \sum_j A_{ij} = \sum_j \omega(i)\Gamma(i \to j)\omega(j). \tag{6}$$

We can also define the transfer matrix $\mathbf{T}$, whose elements $T_{ij}$ are the probability for the system to tunnel from state $i$ to state $j$. This probability is the edge weight $A_{ij}$, multiplied by an overall constant of proportionality chosen such that the total probability of transitioning from any given node is one. We therefore set

$$T_{ij} = \frac{A_{ij}}{d_i} = \frac{\omega(i)\Gamma(i \to j)\omega(j)}{\sum_k \omega(i)\Gamma(i \to k)\omega(k)}. \tag{7}$$

Let $\mathbf{p}(t)$ be a vector whose components $p_i(t)$ are the probability to find the system in state $i$ at a discrete time $t$. For stochastic systems, this probability evolves according to the transfer matrix:

$$\mathbf{p}(t+1) = \mathbf{p}(t)\mathbf{T}. \tag{8}$$

As $t \to \infty$, the system will approach a steady state configuration $\mathbf{p}_\infty$ that is a fixed point of the time evolution such that

$$\mathbf{p}_\infty = \mathbf{p}_\infty \mathbf{T}. \tag{9}$$

That is, $\mathbf{p}_\infty$ is the left eigenvector of $\mathbf{T}$ with an eigenvalue of one. Importantly, $\mathbf{T}$ is a column-stochastic matrix (i.e. each of its columns sums to one), and so all of its eigenvalues are guaranteed to have magnitude $|\lambda| \leq 1$. The *eigenvector centrality* of a matrix is defined to be the left eigenvector with the largest eigenvalue, which means that the eigenvector centrality of the transfer matrix is a left eigenvector with an eigenvalue of one. This is precisely the criterion for $\mathbf{p}_\infty$ to be a fixed point of time evolution in (9), and so $\mathbf{p}_\infty$ is the eigenvector centrality of $\mathbf{T}$. Therefore, by computing the eigenvector centrality of the network, we immediately know what the steady-state configuration of the system is at late times.

An analytic expression for $\mathbf{p}_\infty$ can easily be obtained when the tunneling amplitudes $\Gamma(i \to j)$ are symmetric under exchange of $i$ and $j$, which is the case in our models 1 & 3. Physically, this can be interpreted as considering ensembles of states that are at approximately the same energy and so the tunneling amplitude between any two states is the same in both directions, i.e. no irreversible relaxation processes occur in addition to or in tandem with tunneling processes. The eigenvector centrality of such a network is then given exactly by [46,47]

$$p_{\infty,i} = \frac{d_i}{\sum_k d_k} = \frac{\sum_j \omega(i)\Gamma(i \to j)\omega(j)}{\sum_{k,l} \omega(k)\Gamma(k \to l)\omega(l)}. \tag{10}$$

Notice that this expression is independent of the initial conditions in the network. No matter which state (or what probabilistic superposition of states) the system begins in, it always evolves to a late-time steady state given by (10), which only uses information about the degeneracies of each state and the transition amplitudes between states.

## 2.2 Random Walks on Networks

One problem with using the analytic result (10) for the network centrality is that it requires computing the degree of every node on the network. For very large networks this kind of computation can be computationally expensive and unfeasible to do. Another issue is that the centrality only tells you the behavior of the system in the $t \to \infty$ limit; it doesn't capture any of the finite-time behavior of the network. In these cases, we can instead understand the evolution of the system by performing a random walk on the network.[7]

In a random walk, if at a discrete time $t$ the system is at node $i$, then at the next time step $t+1$ the system moves randomly to a neighboring node according to the probabilities in the transfer matrix $\mathbf{T}$. Once we have done a sufficiently large number of such time iterations, we can tally up what fraction $f_i$ of steps were spent in node $i$. If the random walk were to run for an infinite amount of time, the fraction $f_i$ would converge precisely to the steady-state probability $p_{\infty,i}$. For finite-time random walks, the fraction $f_i$ will serve as a good estimate of $p_{\infty,i}$, as long as the random walk has run for longer than the characteristic relaxation time of the network [46, 48]. For a more detailed discussion of random walk convergence, see appendix A.

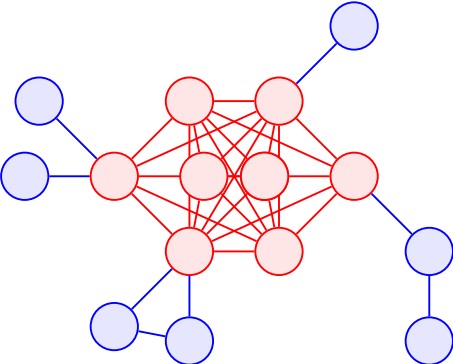

Figure 4: A network with a highly-connected subnetwork, indicated in red. Random walks on such networks typically do not require traversing the entire network.

Random walks are often more efficient to compute than the actual centrality because they rely on only local neighborhoods of nodes and not the full network. For example, consider the network depicted in figure 4, where the highly-connected nodes only comprise a small subnetwork of the full network. Performing a random walk on such a network will typically only require computing the transfer matrix elements on the highly-connected subnetwork, whereas the centrality (10) requires computing all entries of the transfer matrix.

## 2.3 Other Network Properties

We have so far only discussed methods for determining the late-time behavior of a quantum mechanical system. However, this only scratches the surface of the wide array of network-theoretic tools that can be used to gain insight into non-trivial properties of quantum systems. For example, community detection algorithms can be used to look for the presence of highly-connected subnetworks, which can be thought of as subspaces of the full Hilbert space whose dynamics are approximated by truncating the full Hilbert space onto the subspace [49–52]. Additionally, more refined versions of the eigenvector centrality can be constructed by modifying the transfer matrix in particular ways; these generalized eigenvector centralities can give insight into late-time behavior when features like damping, sources and sinks, and random

---

[7]For a good review of random walks on networks, see e.g. [30, 46].

noise are present [53,54]. All in all, we believe that this network-based approach to quantum mechanics is a fruitful topic to explore for a wide range of physical systems.

# 3   Model 1: $N$ Centers

## 3.1   Setup

In our first model, as discussed in section 1.1, we will model multi-centered black hole microstates very minimally. Every microstate in our model will be imbued with only one property: the number $N$ of centers that it has. We will set a cut-off on how many centers a microstate can have by demanding that $1 \leq N \leq N_{\max}$ for some maximum number of centers $N_{\max}$.

We want to associate a degeneracy to each microstate with $N$ centers, related to the number of ways to add a slight non-extremality excitation to the given microstate (as also discussed in section 1.1). Our model for the degeneracy of black hole microstates is[8]

$$\omega(N) = N^{\beta}, \tag{11}$$

where $\beta$ is a tuneable numerical parameter. If the most important contribution to the degeneracy function comes from the number of ways to "wiggle" bubbles in a multicentered solution, larger bubbles should give a larger degeneracy; since larger bubbles typically arise when there are fewer centers, we would then expect $\beta \leq 0$. On the other hand, if the most important contribution to the degeneracy comes from the configurational entropy of the $N$ centers (i.e. rearranging them in space) or from adding small velocities to the centers, then a larger number of centers would give a larger degeneracy and we would expect $\beta \geq 0$. We will consider both situations for the parameter $\beta$.

We model the transition rate between two microstate solutions by

$$\Gamma(N \to N') = \exp\left(-\gamma \min(N, N')^{\delta}\right), \qquad \text{for } |N - N'| \leq 1, \tag{12}$$

where $\gamma$ and $\delta$ are some numerical parameters. Quantum tunneling rates are exponentially suppressed, so it is natural to demand $\gamma \geq 0$. We also expect that it should be easier for tunneling to occur when there are more centers present, since then the bubbles are smaller and thus give rise to lower potential barriers. (This is also congruent with the results of [27], which found a higher tunneling rate for a larger number of centers.) We therefore will choose $\delta \leq 0$ in order for the tunneling rate to be suppressed for small $N$. Note also that we take the minimum of $N$ and $N'$ in order to guarantee that the tunneling rate is symmetric when tunneling between $N$ and $N'$. Importantly, the only transitions allowed are $N \to N' = N, N \pm 1$.

Since the only property of the microstates we are looking at is the number of centers, any two microstates with the same number of centers will appear identical. So, the probability of going from an $N$-center microstate to any microstate with $N'$ centers must be weighted by the degeneracies $\omega(N)$, $\omega(N')$ of the initial and final microstates. The probability $P(N \to N')$ to tunnel from a microstate with $N$ centers to one with $N'$ centers is therefore given by

$$P(N \to N') = \frac{\omega(N)\Gamma(N \to N')\omega(N')}{\displaystyle\sum_{n} \omega(N)\Gamma(N \to n)\omega(n)}, \tag{13}$$

where the normalization is chosen such that all probabilities sum to one. Note also that any constant prefactors in the degeneracies and transition rates will cancel in this expression; it

---

[8]We have also considered an exponential degeneracy function of the form $\omega(N) = \exp\left(\gamma' N^{\beta'}\right)$ for $\gamma' = \pm 1$ and $\beta' \in (-1, 1)$. As we discuss in section 3.3, this functional form of the degeneracy function or (11) gives the same qualitative results (when $\beta, \beta'$ have the same sign).

is only the relative differences in degeneracies and transition rates that affect the late-time behavior.

The dynamics of this model can be captured by the simple network shown in figure 5. There are $N_{\text{max}}$ nodes in the network, each labeled by the number of centers of the microstates they describe. The edges are directed and represent allowed transitions in the model. The weight of each edge is the corresponding probability for that transition to occur. The adjacency

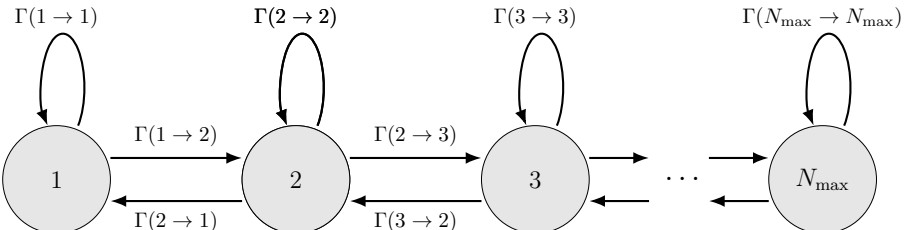

Figure 5: A network representation for model 1. Each node corresponds to a different value of the number of centers $N$ and each (directed) edge is weighted by the probability to tunnel from one value of $N$ to another.

matrix $\mathbf{A}$ of this network has elements $A_{ij} = \omega(i)\Gamma(i \to j)\omega(j)$, while the transfer matrix $\mathbf{T}$ has elements $T_{ij} = P(i \to j)$ (using (13)). Note that $i$ and $j$ range from 1 to $N_{\text{max}}$, so $\mathbf{A}$ and $\mathbf{T}$ are $N_{\text{max}} \times N_{\text{max}}$ square matrices. Moreover, their definitions here are consistent with the general formulas presented in section 2.

The eigenvector centrality $\mathbf{p}_\infty$ (i.e. the left eigenvector of $\mathbf{T}$ with an eigenvalue of one) determines the late-time behavior of the system. In particular, the late-time probability of being in a node with $N$ centers is simply the $N^{\text{th}}$ component of $\mathbf{p}_\infty$. We can also compute the expected value $\langle N \rangle$ of the number of centers at late times via

$$\langle N \rangle = \sum_n n\, p_{\infty,n}. \tag{14}$$

## 3.2 Results

Now that we have established the setup of our model, we now want to explore how the numerical parameters of the model affect the late-time behavior of the black hole microstates. Specifically, we will look at how the parameters affect the eigenvector centrality of the our network model and make conclusions about what kind of microstate we expect to be in at late times. We will consider the effects of the degeneracy parameter $\beta$ introduced in (11) and the transition rate parameters $\gamma, \delta$ introduced in (12). Note that, for this simple network, we can simply calculate the analytic and exact late-time probability vector as given in (10), and thus do not need to actually perform explicit random walks on this network.

### 3.2.1 Degeneracy Dependence

We first want to analyze how the degeneracy $\omega(N)$ of microstates affects the eigenvector centrality. A plot of the degeneracy versus the number of centers for various values of $\beta$ is given in figure 6. When $\beta = 0$, the degeneracy is uniform and there are an equal number of microstates for any value of $N$. As $\beta$ is tuned below zero, though, the degeneracy is slanted towards microstates with a small number $N$ of centers. We would therefore expect that setting $\beta$ close to zero makes the eigenvector centrality uniform, since there are an equal number of states for all values of $N$, while tuning $\beta$ to be more negative corresponds to shifting the centrality towards smaller values of $N$. Making $\beta$ positive would also simply push the centrality towards larger values for $N$. Our intuition is confirmed by the eigenvector centrality of our system; a plot of

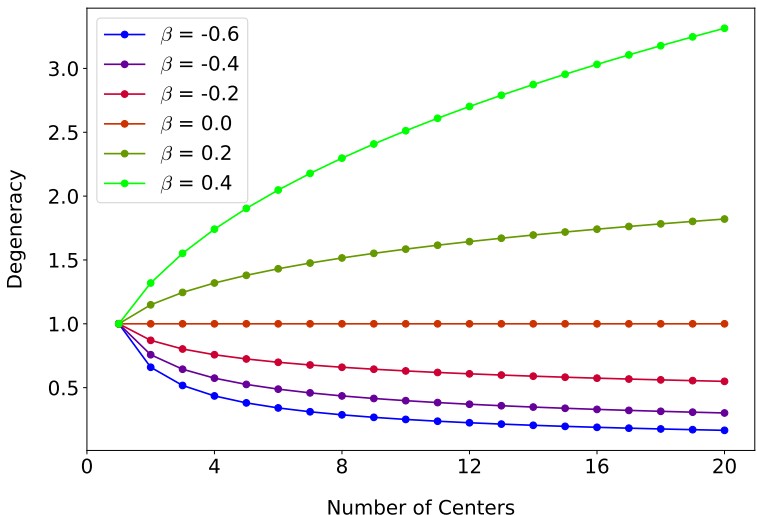

Figure 6: The degeneracy $\omega(N)$ versus $N$ for a range of values of $\beta$, with $N_{\max} = 20$.

the eigenvector centrality for multiple values of $\beta$ and $N_{\max} = 20$ are shown in figure 7, for now setting $\delta = 0$ (and $\gamma = 1$). We see that we can smoothly tune the eigenvector centrality to be pushed entirely to small values of $N$ by making $\beta$ more negative.

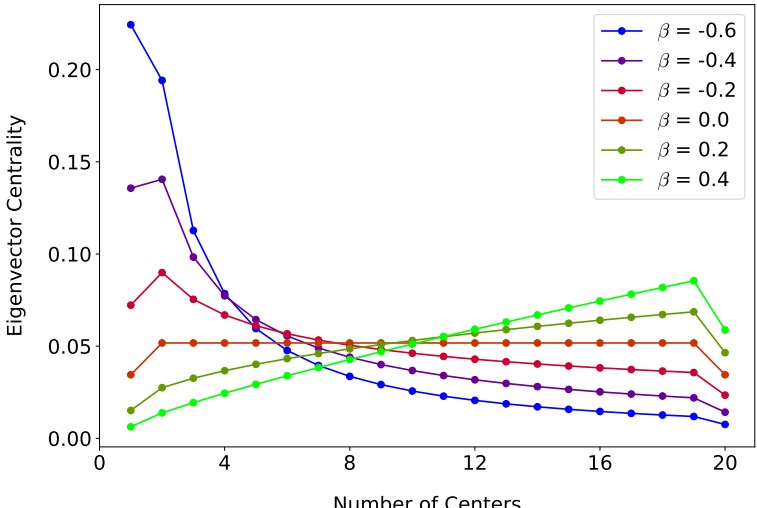

Figure 7: The eigenvector centrality for different values of $\beta$ when $\delta = 0$ and $N_{\max} = 20$.

This trend persists when $\delta \neq 0$ as well (still with $\gamma = 1$). Heat plots of $\langle N \rangle$ for a range of $\beta$ and $\delta$ values are given in figure 8.[9] No matter what value of $\delta$, $\gamma$, and $N_{\max}$ we look at, very negative values of $\beta$ push $\langle N \rangle$ close to one while values of $\beta$ close to zero push $\langle N \rangle$ close to $N_{\max}/2$. Moreover, the crossover between these two regions is smooth and continuous, with no sharp phase transitions appearing.

---

[9]We only plot negative values of $\beta$ in the heat plots; for negative values of $\beta$, the (blue) trend of the heat plots of fig. 8 simply continues on to the left.

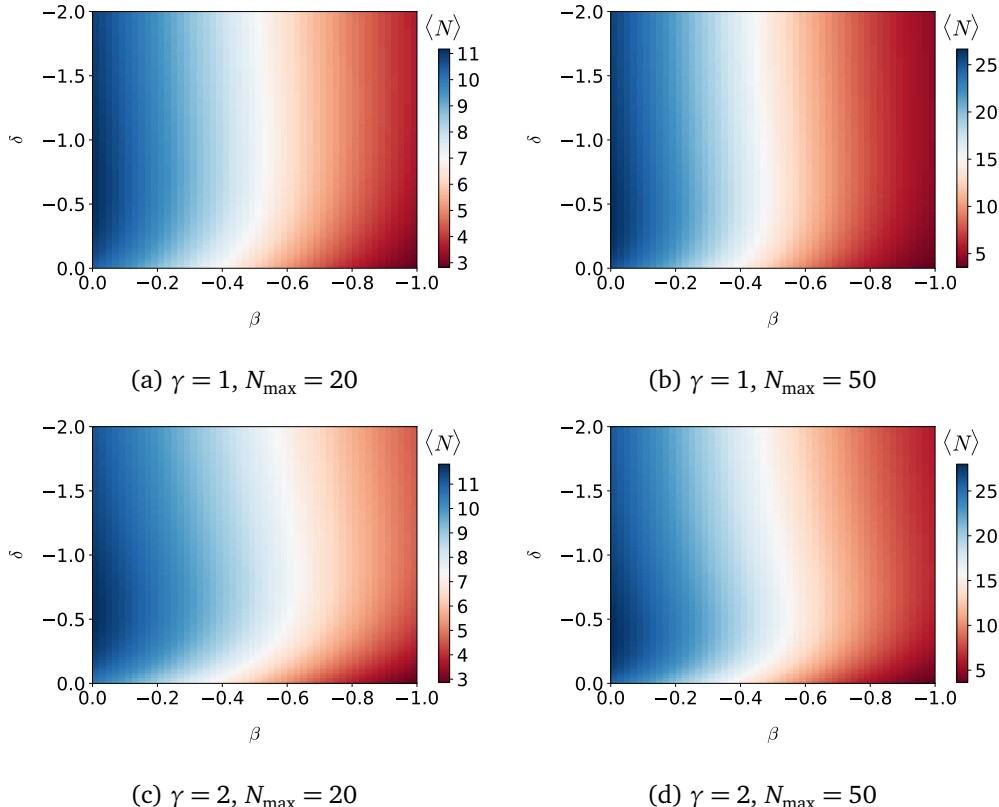

(a) $\gamma = 1$, $N_{\max} = 20$

(b) $\gamma = 1$, $N_{\max} = 50$

(c) $\gamma = 2$, $N_{\max} = 20$

(d) $\gamma = 2$, $N_{\max} = 50$

Figure 8: Heat plots showing $\langle N \rangle$ for a range of $\beta$ and $\delta$ values. No matter what values of $\gamma$ and $N_{\max}$ are chosen, the $\delta$ and $\beta$ dependence remains roughly the same.

### 3.2.2 Transition Rate Dependence

One immediately striking fact about the centrality results given in figure 8 is that the transition rate is much less impactful than the degeneracy in determining the centrality of the network. Nonetheless, the transition rate still affects the centrality in a non-trivial way.

The functional form is the same for all three transitions, so for the purpose of understanding the transition rate we will first just look at the $N \rightarrow N + 1$ transition. Plots of the transition rate $\Gamma(N \rightarrow N + 1)$ versus $N$ for various values of $\delta$ are shown in figure 9. When $\delta = 0$, the transition rate becomes independent of $N$ and $N'$ and thus uniform. The centrality is therefore determined entirely by $\beta$ when $\delta = 0$. As $\delta$ is first tuned below zero, the transition rate becomes non-uniform; instead, it increases with $N$. This means that transitions are more likely between microstates with higher numbers of centers, and so we would expect the centrality to be shifted towards higher values of $N$. As we continue to tune $\delta$ below zero, though, this effect becomes less pronounced; the transition rate is mostly uniform in $N$ for very negative values of $\delta$. For very negative values of $\delta$, then, the centrality is once again determined entirely by $\beta$. We can thus conclude that $\delta$ should push the system to larger values of $\langle N \rangle$ when $\delta$ is negative, though this effect should diminish as $\delta$ becomes very negative. Again, our intuitive picture is confirmed by plotting the centrality for $\beta = 0$ and varying $\delta$ in figure 10.

We can also look at how $\langle N \rangle$ varies with $\delta$ for different values of $\beta$. Some plots of this are shown in figure 11. These plots again demonstrate precisely the behavior we expected from our analysis of the transition rates plotted in figure 9.

The behavior we have observed above is generic, in the sense that it holds for any value of $\beta$ or $\gamma$. The effect $\gamma$ has is to make the peak in the $\langle N \rangle$ versus $\delta$ plots more pronounced. As we can see from figure 11, when $\gamma$ is tuned below zero the peaks become sharper and more

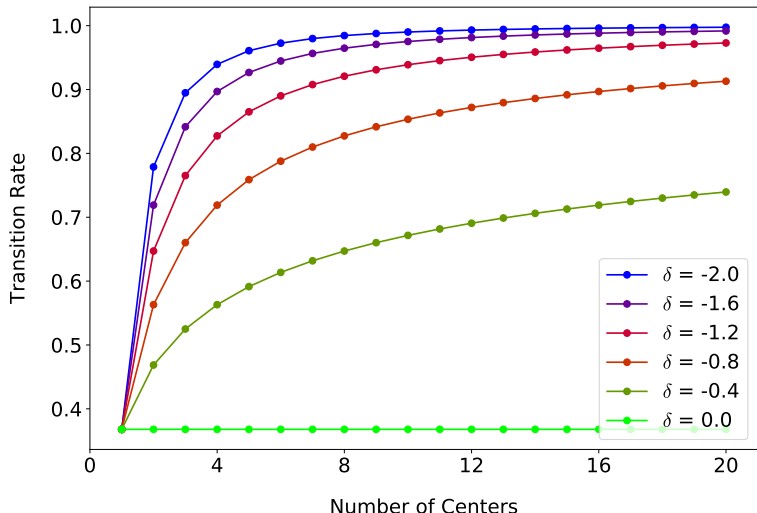

Figure 9: The transition rate $\Gamma(N \to N+1)$ versus $N$ for a range of values of $\delta$, with $\gamma = 1$.

defined. This can also be seen from the heat map plots in figure 8, where the contours are clearly sharper for the $\gamma = -2$ heat maps than the $\gamma = -1$ heat maps. Once again, though, this effect is small compared to the effect $\beta$ has on the centrality.

## 3.3 Conclusions

The main conclusion we can derive from our analysis of this network is that the late-time behavior of the microstates is dominated by their degeneracy $\omega(N)$ (as opposed to their transition rates $\Gamma(N \to N')$). The eigenvector centrality of the network is primarily determined by the parameter $\beta$ in the degeneracy, with values of $\beta$ close to zero making the centrality uniform in $N$ and more negative (resp. positive) values of $\beta$ pushing the centrality to be larger for microstates with smaller (resp. larger) $N$. The values of $\gamma$ and $\delta$ in the transition rates give rise to small modulation of on top of these effects; in particular, $\delta$ negative (but not too negative) pushes the centrality to be slightly larger for larger values of $N$, while negative values of $\gamma$ makes this $\delta$-effect more relevant.

Another noticeable feature we found is that the physics of the network is "smooth", in the sense that the parameters $\beta$, $\gamma$, and $\delta$ can be tuned continuously with no sudden spikes or jumps in the centrality. In particular, the parameters can be tuned as needed to make the expected value $\langle N \rangle$ at late times whatever value we want. This will not be true in model 2 below.

One could wonder if the degeneracy function $\omega(N)$ could have a different functional dependence on $N$ than we have considered in (11). We have also considered an exponential degeneracy function of the form:

$$\omega(N) = \exp\left(\gamma' N^{\beta'}\right), \tag{15}$$

where we considered $\gamma' = \pm 1$ and $\beta' \in (-1, 1)$. We found that the physics of such a degeneracy function is qualitatively the same as the polynomial degeneracy (11) and thus follows the same qualitative behaviour as already discussed in this section.

In order to interpret the degeneracy dominance of the network, we can recall the interpretation of our network and its parameters. The degeneracy function $\omega(N)$ represents the number of slightly non-extremal $N$-center microstates, while the transition rate $\Gamma(N \to N')$

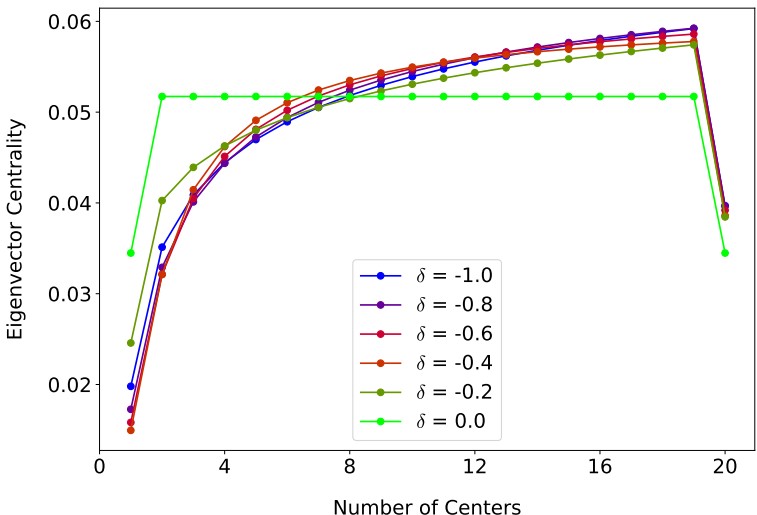

Figure 10: The eigenvector centrality for different values of $\delta$ when $\beta = 0$, $\gamma = 1$, and $N_{\max} = 20$.

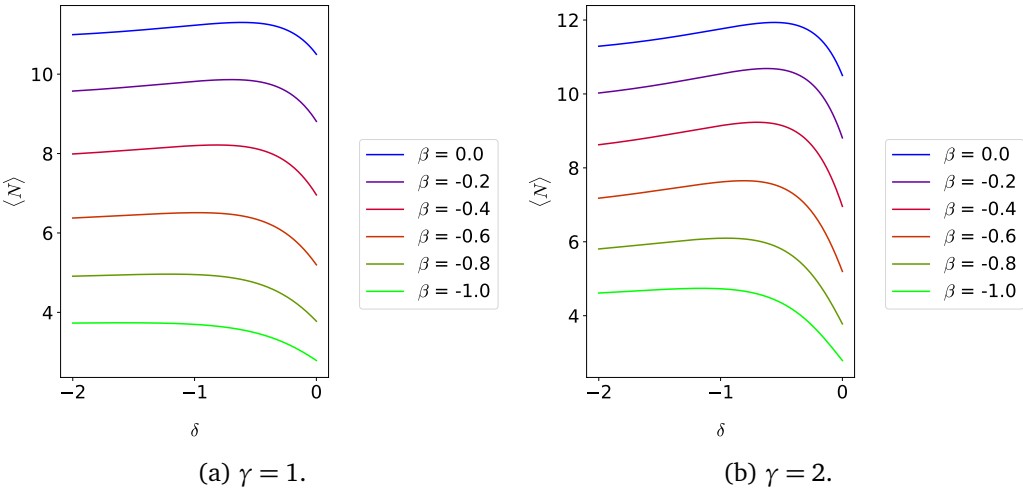

(a) $\gamma = 1$.                              (b) $\gamma = 2$.

Figure 11: Plots of $\langle N \rangle$ versus $\delta$ for different values of $\beta$, with $N_{\max} = 20$.

represents the quantum tunneling probability between such microstates (such as calculated in [27]). With this picture in mind, the degeneracy dominance of our results indicates that *the counting of non-extremal microstates is much more important than the details of the tunneling interactions between the microstates.* As we have mentioned, the counting of such non-extremal microstates depends on the counting of the initial (BPS) microstates as well as the number of ways one can add a slight non-extremality to a given microstate.

We interpret the dynamics of the microstates in this model as being near-BPS states whose collective dynamics are *ergodic,* in the sense that, given enough time, any microstate will eventually tunnel into any other given one. At any given time, the system is (approximately) equally likely to be in any microstate.

# 4 Model 2: $N$ Centers with Charge

## 4.1 Setup

In our second model, we want to add more features to our previous model of multi-centered black hole microstates. As discussed in section 1.1, we will consider microstates where all centers lie on a line, as depicted in fig. 12. We will model this microstate as having a total charge $Q$ that is distributed among each of the centers.[10] Thus, we can view the corresponding microstates as ordered partitions (i.e. *compositions*) of the total charge $Q$; each microstate has a number of centers $N$, as well as a set of charges $\{Q_i\}$ concentrated at each of the centers that sum up to the total charge $Q$. We will also require that each center contains at least one unit of charge and that all charges are positive; this implies the maximum possible number of centers is $N_{\max} = Q$.

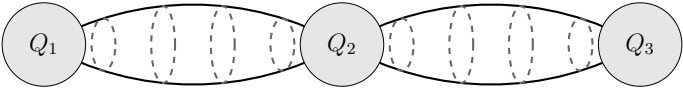

Figure 12: A sample charged black hole microstate. This one has three centers, each with an associated charge, as well as topological bubbles stretched between each adjacent center.

Like we did with model 1 (and as discussed in section 1.1), we model the possible degeneracy of adding non-extremality to a microstate with the degeneracy function:

$$\omega(N, \{Q_i\}) = \prod_{i=1}^{N} \alpha Q_i^{\beta}, \tag{16}$$

where $\alpha$ and $\beta$ are some numerical parameters. If this degeneracy is dominated by the number of ways to excite or "wiggle" the topological bubbles between centers, then the degeneracy should be greater for microstates with larger bubbles (and thus larger charges), and so we would expect $\beta \geq 0$. On the other hand, if the degeneracy is dominated by the number of ways to add small velocities to the centers, then the microstates with a large number of centers (and thus smaller charges) will be more degenerate, which requires $\beta \leq 0$. To cover both cases, we will consider both cases for $\beta$.

The transitions between microstates in this model can be pictured as breaking off an amount of charge from a certain center and tunneling it onto an adjacent center or tunneling it to create a brand new (adjacent) center. (We will not allow charge to tunnel elsewhere, i.e. we will not allow charge to "hop" over existing centers.) Our model for the tunneling rate is:

$$\Gamma\left(N, \{Q_i\} \to N', \{Q_i'\}\right) = \exp\left(-\gamma Q_T^{\delta} Q_L^{\lambda} Q_R^{\lambda}\right), \tag{17}$$

for some tunable parameters $\gamma$, $\delta$, and $\lambda$. $Q_T$ is the amount of charge that has broken off the original center and tunnels away. $Q_{L,R}$ are measures of how much charge sits to the left and right, respectively, of the center that the charge tunnels from. If the charge tunnels away from

---

[10]The fact that each center contributes a well-defined, separatable amount to the total asymptotic charge is necessarily an oversimplifying assumption of our model that is not quite correct in an actual Bena-Warner multi-centered solution. For example, in appendix B.2, we derive the formulae (64)-(66) and (67)-(68); these are explicit examples showing that the contribution to the total asymptotic charges of e.g. individual supertubes cannot be separated entirely — there are always "cross-terms" between different supertubes in the expressions for the asymptotic charges. Note that also the assumption that all centers have positive charge is an oversimplification; centers are allowed to have negative charge in actual multicentered solutions.

the $i^{\text{th}}$ center, these are given by

$$Q_L = \max(\tilde{Q}_L, 1), \quad \tilde{Q}_L = Q_{i-1} + \omega Q_{i-2} + \omega^2 Q_{i-3} + \ldots = \sum_{j=1}^{i-1} \omega^{j-1} Q_{i-j},$$

$$Q_R = \max(\tilde{Q}_R, 1), \quad \tilde{Q}_R = Q_{i+1} + \omega Q_{i+2} + \omega^2 Q_{i+3} + \ldots = \sum_{j=1}^{N-i} \omega^{j-1} Q_{i+j}, \tag{18}$$

where $\omega$ is a parameter that encodes how far-ranging the electromagnetic force is. We require that $0 \le \omega \le 1$, which ensures that centers closer by will have a larger effect on the tunneling rate. Since $Q_L, Q_R$ only depend on the initial (and not final) microstate in the tunneling process, the tunneling rate (17) is not symmetric under interchange of initial and final states. Note also that if there are multiple transitions possible from one microstate to another, then we take all of them into account simultaneously — the total tunneling rate is then simply the sum over the tunneling rates of all these possible transitions.

It is interesting to make contact between our transition rate (17) and the explicit tunneling rates calculated in specific multicentered microstates in [27]. Of course, our simplistic model does not capture the intricacies of the actual multi-center microstate solutions. Nevertheless, from [27], it is clear that the physical choice for the parameter $\delta$ is $\delta = 1$. In appendix B.3, we make an attempt to also extract very rough estimates for the values of $\lambda$ and $\omega$ from explicit microstate tunneling calculations; the result is $\lambda \approx -0.18$ and $\omega \approx 0.37$. Although these values for the parameters could be argued to be the most physically relevant, we will still consider varying these parameters in order to explore the full parameter phase space of our model. Finally, we note that [27] assumes (but does not explicitly construct or verify) that it is possible to construct many microstates with different numbers of centers $N$ that have the same asymptotic charges at infinity[11]. In appendix B.2, we give a proof of concept that it is possible to "split" a center into multiple centers while keeping *all* asymptotic charges fixed by constructing an explicit example of such a splitting.

The dynamics of model 2 can be encoded into a network where each node is represented by an ordered partition (i.e. composition) of the total charge, while the edges represent the allowed transitions; an example is shown in figure 13. Importantly, the number of composi-

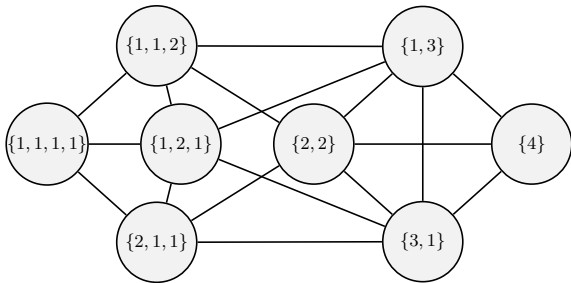

Figure 13: The network in model 2 when the total charge is $Q = 4$. (Unlike depicted in this simplified network figure, note that the transitions between states are not always symmetric, see (17).)

tions of $Q$ is $2^{Q-1}$ and thus scales exponentially with $Q$; generating the entire explicit network for $Q \gtrsim 20$ becomes computationally unfeasible. Instead, we will perform dynamic random walks that only generate nodes as needed on the network as the random walk progresses. We perform the random walk until its behavior has converged to a steady-state behavior; see ap-

---

[11]In fact, especially in the non-scaling solution family of [27], the asymptotic angular momentum is *not* constant for microstates of different $N$.

pendix A for details. Then, as discussed in section 2.2, we can use the fraction of steps spent in each node to numerically estimate the late-time behavior of our model.

Once we fix the model parameters and perform a random walk, there are two main pieces of information we can extract from the random walk after it has converged: how many centers the node had at each time step, and how charge was distributed among these centers. For example, consider the case where $\alpha = \gamma = \delta = 1$, $\beta = \lambda = 0$, and $Q = 20$. One random walk performed with these parameters is given in figure 14.

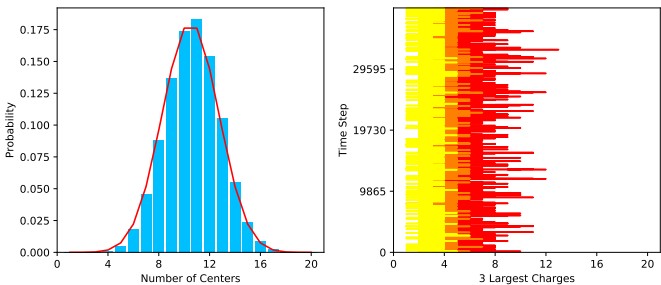

Figure 14: A sample random walk done in model 2. On the left of each plot is the random walk probability to have a particular number of centers, with the fractional number of microstates with a particular number of centers plotted in red. On the right is a plot of the three largest charges present at each step in the random walk, shown in red, orange, and yellow.

On the left in this figure is the probability for the random walk to be in a state with a particular number of centers, plotted in blue. This probability is calculated simply by tabulating the random walk results and counting the fraction of steps spent at each value of $N$. The superimposed red line in this plot is the number of microstates that have a particular number of centers $N$, normalized by the total number of states. The probability distribution will match this exactly when all microstates were equally likely; we will refer to this as *ergodic* behavior (precisely as in model 1), since all nodes in the network will eventually be visited by a random walker with (approximately) equal probability.

On the right of this figure is a plot of the three largest charges present at each time step in the random walk, plotted in red, orange, and yellow (in descending order). These can be useful to determine whether or not a random walk is trapped at a particular node, because we can end up in situations where the number of total centers is unchanging but charges are nonetheless tunneling between the centers.

In the case shown in figure 14, the random walk probability is very close to the number of states as a function of $N$, with an expected value of $\langle N \rangle \approx 11$. In the plot above we have set $\alpha = 1$ and $\beta = 0$, so this red curve is only representing how many compositions of the charge $Q$ have one center, how many have two centers, etc. We can modify this number through the degeneracy function $\omega$ (which depends on $\alpha$ and $\beta$), which models how many additional ways there are to lift each of these states away from extremality.

Additionally, we note from the right-hand side of figure 14 that the three largest charges tend to stay below $Q_i = 8$, with relatively few excursions to very large charges., This matches the suppression of large-charge states in the degeneracy. We can therefore conclude for this set of parameters that the system is demonstrating ergodic behavior; there are no significant departures of the late-time probability distribution from the degeneracy.

Of course, this is just a single example, meant to demonstrate how our random walk results are tabulated. In the next section, we will use many different random walk results to form broad conclusions about our model.

## 4.2 Results

Now that our methodology is clear, we will look at random walk results throughout our parameter space. We will consider the effects of varying the degeneracy parameters $\alpha, \beta$ introduced in (16), and the transition parameters $\gamma, \delta, \lambda, \omega$ introduced in (17) and (18).

We first want to investigate how the late-time behavior depends on the transition rate (17). We will therefore first fix $\alpha = 1$ and $\beta = 0$ in order to set the degeneracy to be unity for all nodes. The new features of this model are the parameters $\lambda$ and $\omega$, which encode how much resistance the tunneled charge experiences from nearby centers. We will first focus on understanding these parameters by fixing $\gamma = \delta = 1$, with $Q = 20$. A number of random walks for different values of $\lambda$ and $\omega$ are shown in 15. For low $\lambda$, the random walks demonstrate mostly ergodic behavior, with the random walk probability matching the degeneracy very well. A caveat to this is that increasing $\omega$ seems to push the true peak slightly to the right of the degeneracy peak. Nonetheless, the behavior of the system is still mostly ergodic.

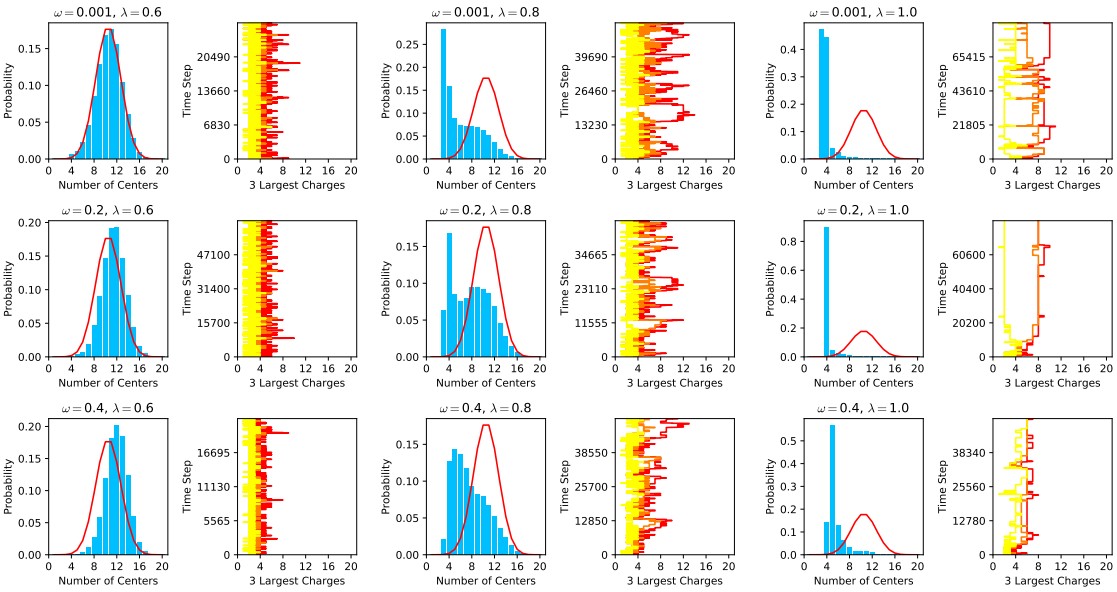

Figure 15: Random walk results for different values of $\omega$ and $\lambda$; other parameters are fixed such that $Q = 20$, $\alpha = 1$, $\beta = 0$, $\gamma = 1$, and $\delta = 1$.

As $\lambda$ increases, though, the system begins to depart drastically from this ergodic behavior. For $\lambda = 0.8$, we can see that the random walk starts to favor microstates with small numbers of centers and larger charges. The plot of the three largest charges is still fluctuating, though, indicating that transitions between these few-center states still occur. At $\lambda = 1$, though, these transitions stop occurring. The random walk very quickly becomes locked in or trapped in a state with around $N = 4$ centers, and very few transitions from that state occur. This indicates that at $\lambda = 1$, the system enters a *trapped phase* with much more rigid and constant behavior than the ergodic phase. Crucially, this seems to be true for all three values of $\omega$ in figure 15.

To investigate this trapped behavior further, we can determine $\langle N \rangle$ as a function of $\lambda$ for a number of different random walks, as depicted in figure 16. In these, we can see that $\langle N \rangle$ is around half of the total charge, as expected for an ergodic phase, for $\lambda < 1$. There is a small increase in $\langle N \rangle$ as $\lambda$ increases in this range, but only a small one (on the order of one center). As $\lambda$ gets very close to 1, though, the system enters the trapped phase and $\langle N \rangle \approx 4$. This critical behavior is robust, in the sense that it is relatively unaffected by changing the values of $\omega$ and $\delta$. Additionally, the plots shown have $Q = 20$, but the same features appear for larger values of $Q$ as well.

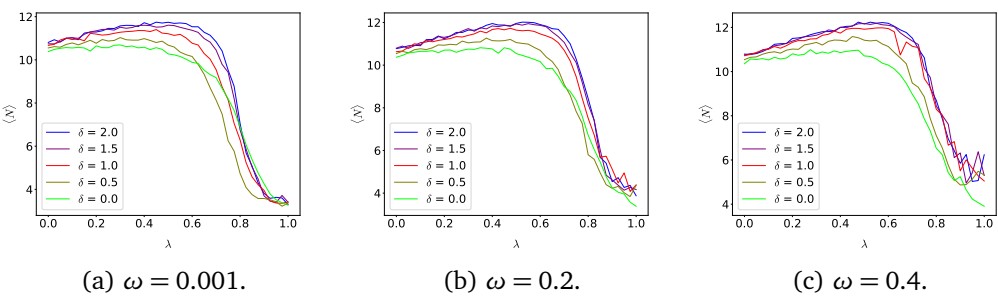

(a) $\omega = 0.001$.     (b) $\omega = 0.2$.     (c) $\omega = 0.4$.

Figure 16: Plots of $\langle N \rangle$ versus $\lambda$ for different values of $\omega$, with $Q = 20$.

We can also consider varying the parameter $\delta$ (which was kept constant at $\delta = 1$ above). It is most convenient to illustrate this with the heat plots shown in figure 17, which give $\langle N \rangle$ for a range of $\lambda$, $\delta$, and $\omega$ values. From these plots, we can immediately see that the phase transition at $\lambda = 1$ is present for any values of $\delta$ and $\omega$. $\delta$ has a very small effect on the results; larger values of $\delta$ push $\langle N \rangle$ to be slightly larger or smaller, depending on if $\lambda > 0$ or $\lambda < 0$, respectively. This is consistent with our results from section 3 where the similar parameter $\delta$ also only provided a small modulation to the late-time behavior.

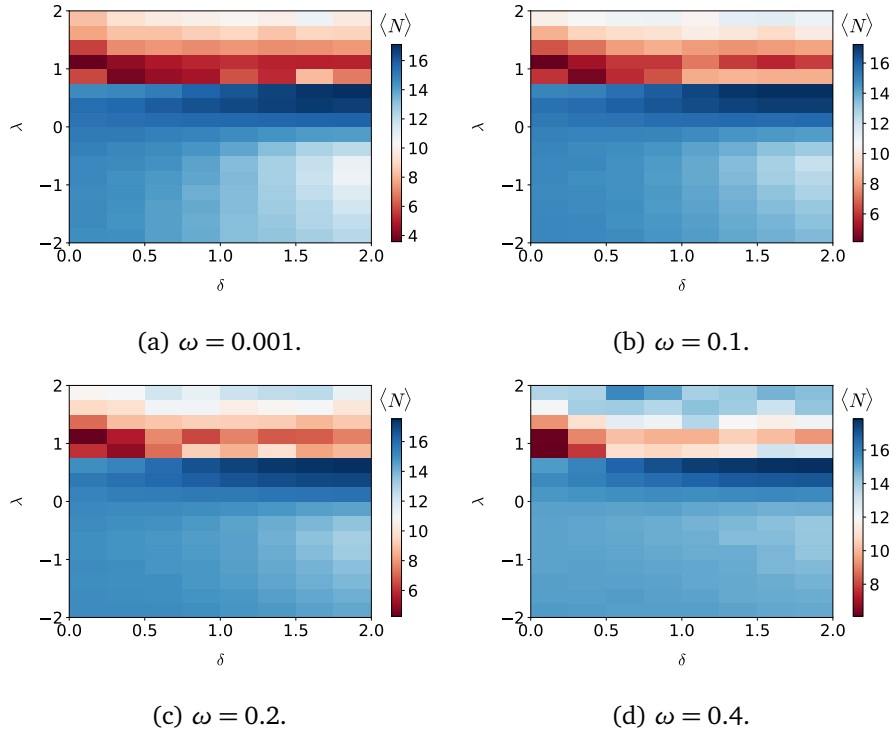

(a) $\omega = 0.001$.            (b) $\omega = 0.1$.

(c) $\omega = 0.2$.            (d) $\omega = 0.4$.

Figure 17: Heat maps of $\langle N \rangle$ as a function of $\lambda$ and $\delta$, for different values of $\omega$, with $\alpha = 1$, $\beta = 0$, $\gamma = 1$, and $Q = 30$.

We have so far only investigated how the late-time behavior of model 2 depends on the transition rate (17). We now need to account for how this depends on the degeneracy (16). Since $\omega \propto \alpha^N$, increasing $\alpha$ will push the random walks to peak at higher values of $N$, while decreasing it pushes the random walk to peak at lower values of $N$. The degeneracy dependence on $\beta$ is a little more complicted; to understand what it does, consider figure 18, where we plot the degeneracy of all microstates with total charge $Q = 20$ and with a particular number of centers $N$ as a function of $N$. The degeneracy has an extremum at intermediate values of

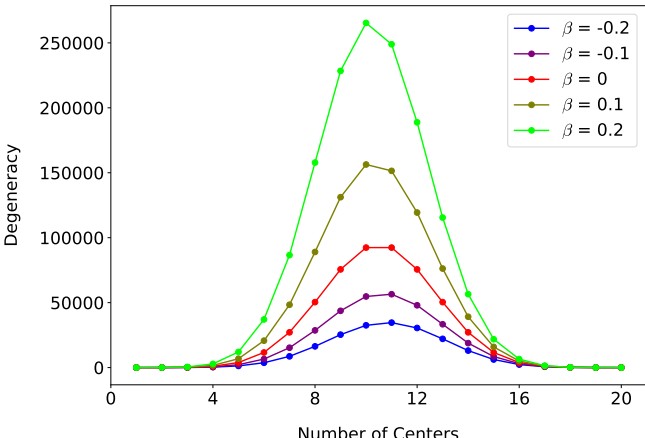

Figure 18: The degeneracy $\omega(N, \{Q_i\})$ versus $N$ for a range of values of $\beta$, with $\alpha = 1$ and $Q = 20$.

$N$; when $\beta < 0$, this peak gets pushed lower and so the intermediate states become relatively disfavored, while for $\beta > 0$ the peak becomes greater and they become relatively more favored. $\beta$ will therefore control the width of the peak in our random walk results, with narrow and wide peaks corresponding to $\beta > 0$ and $\beta < 0$, respectively.

We considered these effects and studied random walk behavior for different values of $\alpha$ and $\beta$, but their only effect was to alter the random walk probability in smooth, continuous behavior similar to what we saw from degeneracy effects in section 3. That is, the parameters $\alpha$ and $\beta$ can be smoothly tuned to shift the location and width of the degeneracy peak as desired. No matter what we set $\alpha$ and $\beta$ to, though, it is the transition rate parameter $\lambda$ that affects how closely the actual centrality matches the ergodic prediction. We will discuss the physics of the $\lambda$-dependence in more detail in the next two subsections.

### 4.2.1 Phase Transition

The most distinct feature in our results is an apparent phase transition, where the system goes from an ergodic phase with $\langle N \rangle \sim Q/2$ to a trapped phase with $\langle N \rangle \approx 4$ as soon as $\lambda \gtrsim 1$. Intuitively, it is straightforward to see that there should be these two types of phases. For $|\lambda| \ll 1$, the transition rate are largely independent of $Q_L$ and $Q_R$, so the transition rates between states are approximately uniform. This leads us to a distribution where $\langle N \rangle$ is related primarily to the degeneracy of the system. On the other hand, when $\lambda \gg 1$, it becomes much harder for charge to tunnel off of centers if they are close to an area of large charge concentration, as depicted in figure 19. At large $\lambda$, these highly-charged areas act as charge sinks, as charge can easily tunnel onto the sink but is very unlikely to tunnel off of it. The long-time behavior of the system is to end up in microstates with very few total centers. Note that the larger $\lambda$ is, the larger the asymmetry is of the transition rates between the initial and final states; this asymmetry could be interpreted physically as an additional irreversible relaxation process that happens immediately after the (reversible) tunneling process, leading to a total transition rate that is asymmetric between the initial and final states.

What is surprising, though, is how sudden the transition from ergodic to trapped behavior is in parameter space. Instead of having a smooth, gentle transition from one phase to another, the transition is sharp and sudden. Moreover, the random walk numerics are stable in the crossover regime, indicating that this is not simply a numerical artifact arising in our methodology. It would be interesting to investigate this feature further, although (as we have stressed before)

Figure 19: A cartoon of how $\lambda \gg 1$ affects tunneling such that typical microstates have only a few centers.

the large size of our networks make analytic analyses difficult.

One interesting thing to note about the trapped phase of our system is that the network is not locked into one particular microstate. Explicit random walk results (see e.g. figure 15) show that the number of centers and amount of charge on each center fluctuate, but much less than the random walk fluctuations in the ergodic phase. Statistical fluctuations are effectively restricted such that tunneling only occurs among the centers with $\langle N \rangle \approx 3-4$ rather than the whole network; we can effectively truncate our full network to just this subnetwork when analyzing dynamics in the trapped phase.

The phase transition behavior we observe is present across all of parameter space in our model as long as $\omega$ is in the range $0.001 \leq \omega \leq 0.6$. For $\omega > 0.6$, the system becomes stuck; the tunneling rate suppression due to adjacent charges becomes so large that the random walk state is trapped at whatever its initial microstate is. This should not be interpreted as a physical effect. Instead, it indicates that the spectrum of the transfer matrix is highly degenerate such that the network relaxation time is very large; as such, finite-time random walks cease to be a good approximation of the late-time behavior of our network. It seems likely that the eigenvector centrality will still give trapped behavior in this regime, although verifying this is computationally difficult.

### 4.2.2 Smaller $\lambda$ Behavior

Another interesting feature of our results is that, as long as $\lambda$ is below the critical value of the phase transition, the expected number of centers increases monotonically with $\lambda$. This effect is mild (see e.g. figure 16 where $\langle N \rangle$ increases by one or two on the range $0 \leq \lambda \leq \lambda_c$), but it is nonetheless persistent when varying the other transition rate parameters. This indicates that there is some variance possible in $\langle N \rangle$ in the ergodic phase of the system; it can be tuned slightly away from $Q/2$ by small changes in the transition rate.

It is important to note that the intuitive picture we used in section 4.2.1 to explain the phase transition predicts that $\langle N \rangle$ should decrease monotonically with $\lambda$; from that perspective, this observed (opposite) behavior at small $\lambda$ is unexpected. In addition, since this feature is present only in the ergodic phase, it seems likely that one cannot come up with a physical justification for this effect on a truncated subspace of our network. This feature therefore serves as good example of how our network-theoretic approach can lead to emergent phenomena that only become apparent when we consider the dynamics of all states in the theory at once.

### 4.3 Conclusions

Our main result in model 2 is that there is an apparent phase transition in parameter space of the late-time behavior of the microstates. This phase transition is intimately related to the tunneling rates between the microstates, and occurs when the tunneling rate parameter $\lambda$ hits the critical value $\lambda_c \approx 1$. For $\lambda < \lambda_c$, the system is in an *ergodic phase* (similar to that of model 1): the degeneracy of the microstates is far more important than the interactions between them; the system is equally likely to be in any microstate at any given time; and the

model parameters can be tuned to change late-time behavior in a smooth, continuous way. For $\lambda > \lambda_c$, the system is in a *trapped phase* where the late-time behavior is completely dominated by microstates with very few centers and no excursions to other microstates are allowed. This trapped phase demonstrates a regime in parameter space where the details of the transition rates (and not the degeneracy) determine the late-time behavior of the network. We found no such phase in model 1; it is only by adding in more intricate details of charge interactions between centers that this phase appears.

The physical interpretation of these results is as follows. In model 2, we have considered charged microstates whose centers are distributed along a line. When the electromagnetic interactions between the centers are weak, it is easy for charges to move between the centers, and so it is very easy for microstates to tunnel into one another. The system's dynamics are thus ergodic, and all microstates are equally likely to occur. However, if the electromagnetic interactions are sufficiently strong, the microstates with very few centers suddenly become bound states that are very unlikely to tunnel off charge. These microstates therefore become long-lived and metastable, breaking the ergodicity of the system and dominating the time evolution of the black hole microstate system. See also sec. 6 for a discussion relating this behavior to meta-stable black hole glassy physics.

# 5 Model 3: The D1-D5 System

## 5.1 Setup

In model 3, we will model the dynamics of the D1-D5 system with $N_1$ D1-branes and $N_5$ D5-branes, as discussed in section 1.1. We fix $N$, where $N = N_1 N_5$ is the product of the number of D1- and D5-branes. A microstate in this model will be completely given by an unordered collection of integers $\{w_1, w_2, \ldots\}$ with $\sum_i w_i = N$, i.e. each microstate corresponds to a *partition* of the integer $N$. Each integer $w_i$ in the partition can be thought of a string with winding number $w_i$ in the string gas picture, where we have in mind the picture where a ground state in the D1-D5 system can be seen as a gas of strings with total winding number $N$. (For more details, see section 1.1.)

If there are $N_w$ strings with winding number $w$ present in a given microstate, then the degeneracy $\Omega$ associated to each winding number is the number of ways to divide the strings into bosonic and fermionic modes (4), which we repeat here for clarity:

$$
\begin{aligned}
\Omega(N_w) &= \sum_{l=0}^{8} \binom{8}{l} \binom{N_w - l + 7}{7} \\
&= \frac{16}{315}(N_w)^7 + \frac{64}{45}(N_w)^5 + \frac{352}{45}(N_w)^3 + \frac{704}{105}(N_w).
\end{aligned}
\tag{19}
$$

The total degeneracy associated to the microstate is then the product of all such winding number degeneracies:

$$
\omega(\{w_1, w_2, \ldots\}) = \prod_{w | N_w \neq 0} \Omega(N_w).
\tag{20}
$$

As opposed to models 1 and 2, we will *not* imbue an extra degeneracy factor to microstates to account for the possible degeneracy associated to adding a small amount of non-extremality to the microstate; such a generalization could certainly be done in a future iteration of the model.

Transitions are allowed between microstates where a long string with winding $w$ splits into two smaller strings with windings $w_a, w_b$ (with $w = w_a + w_b$), as well as the reverse transition

where two smaller strings with windings $w_a, w_b$ combine into one larger string with winding $w = w_a + w_b$. We model the transition rate between these states as:

$$\Gamma(\{w_1, w_2, \ldots\} \to \{w'_1, w'_2, \ldots\}) = \exp\left(-\gamma\, w^\delta\, [\min(w_a, w_b)]^{\lambda_1}\, [\max(w_a, w_b)]^{\lambda_2}\right), \quad (21)$$

with tuneable parameters $\gamma, \delta, \lambda_1, \lambda_2$. Note that this transition rate includes both the splitting transition (where $w \in \{w_1, w_2, \ldots\}$ and $w_a, w_b \in \{w'_1, w'_2, \ldots\}$) and the combining transition (where $w \in \{w'_1, w'_2, \ldots\}$ and $w_a, w_b \in \{w_1, w_2, \ldots\}$), and is thus manifestly symmetric between these two processes.

We will encode the details of this model into a network where each node is an unordered set of winding numbers that total to $N$, and the edges represent the allowed splitting and combining transitions. One such example with $N = 5$ is shown in figure 20. The number of

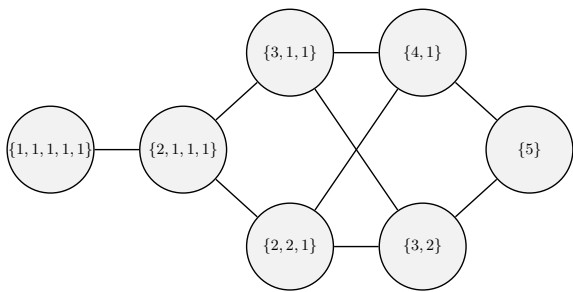

Figure 20: The network in model 3 when the total winding number is $N = 5$.

nodes in the network is simply the number of partitions of the integer $N$, which is known to asymptote to

$$p(N) \sim \frac{1}{4N\sqrt{3}} \exp\left(\pi\sqrt{2N/3}\right), \quad \text{for } N \gg 1. \quad (22)$$

This exponential growth of nodes means that generating the whole network explicitly is computationally unfeasible for $N \gtrsim 10$. So, just as we did with model 2, we will perform dynamic random walks that generate nodes as needed until the random walk has converged to a steady state. The fraction of steps the random walk spends at each node will then serve as a numerical estimate for the late-time behavior of the system, as discussed in appendix A.

## 5.2 Results

In models 1 and 2, we concerned ourselves mainly with discussing the (late-time) expected value of the number of centers. An analogous quantity that we can consider in model 3 is the total number of distinct strings $N_s$, given by the sum over the number of strings per winding number

$$N_s = \sum_w N_w, \quad (23)$$

where we remember that $\sum_w w N_w = N$ is kept fixed. Additionally, in model 2 we investigated the evolution of the centers with the largest charges; here, we will analgously consider the evolution of the strings with the largest winding numbers.

We can now perform random walks and make plots of individual random walk results for different values of the parameters $\delta, \lambda_1, \lambda_2$ in the transition rate (21). A sample of these random walk results are shown in figure 21. These random walks seem to suggest that the actual values of the parameters $\delta, \lambda_1, \lambda_2$ do not actually influence the resulting graph for $N_s$ very much — at most, the peak of $N_s$ can be shifted a slight amount, on the order of $\Delta N_s \approx 1-2$. The (three) largest string sizes also do not seem to depend on the parameter values; they each stay bounded above by $N/2$.

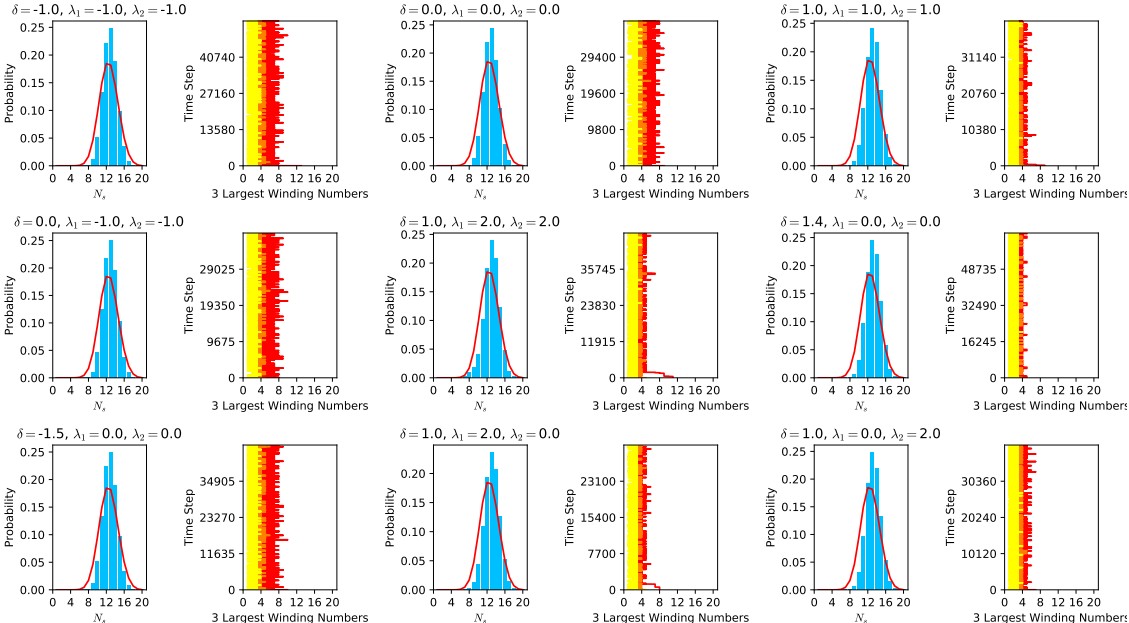

Figure 21: Random walk results for different values of $\delta, \lambda_1, \lambda_2$. On the left of each plot are the probabilities: the random walk probability to be in a microstate with $N_s$ strings is plotted in blue, while the fractional degeneracy of microstates is in red. On the right of each plot we display the three largest winding numbers of the strings in red, orange, and yellow in descending order, respectively, at each time step in the random walk. In all graphs, $N = 20$ and $\gamma = 1$.

We have confirmed this independence of the random walk results on the parameters with a more thorough investigation, exploring the entire space of varying the parameters $\delta, \lambda_1, \lambda_2$ between $-2$ and $2$.[12] We have also tried fixing different values of $N$ and $\gamma$, but the random walk behavior is qualitatively the same as for the $N = 20$, $\gamma = 1$ case shown above. In other words, the functional form of the transition rate (21) does not seem to matter much for the dynamics of our model. In principle, we could have even chosen an entirely different functional form for our transition rate, and we would still see the same random walk peak centered right at the degeneracy peak.

The speed of the convergence of the random walk does depend somewhat on the parameter values. For example, if we increase the parameter $\delta$ to be very large (e.g. $\delta \gtrsim 2$), then it takes a long time for large strings to split into smaller strings. However, once they have split into smaller strings, they will not recombine again as it is much easier for the smaller strings to split into even smaller strings, and so on. The result is that, for such large values of $\delta$, the random walks converge much slower, but they still will converge in the end to approximately the same graphs as those depicted in figure 21.

A second phenomenon that we notice in the graphs of figure 21 is that the actual graph of the number of strings $N_s$ follows the degeneracy function (the red line) but does not exactly match it; rather, the graph's peak is taller and narrower than that of the degeneracy function. This is a non-trivial feature which is indicative that the structure of the network in this model is such that the states around the degeneracy peak are *highly connected*. In fact, these states are so highly connected that, although a random walker can go to any node in the network, they are much more likely to be at these states than at states with much lower or higher values

---

[12]Beyond this range, the random walk runs into numerical convergence problems. Nonetheless, the results that we have looked at beyond this range demonstrate the exact same parameter-independence.

of $N_s$. This results in the network centrality looking like an amplified version of the degeneracy peak. We will call this behaviour the *amplified phase*, to distinguish it from what we called the *ergodic phase* in models 1 and 2, where the centrality graph exactly followed the degeneracy graph (see e.g. figure 14) and all states were approximately equally likely. In the amplified phase we are seeing here in model 3, individual states that sit at the degeneracy peak are actually *more likely* than other individual states (as opposed to the ergodic phase).

## 5.3  Conclusion

Our main result from this section is that model 3 exhibits late-time behavior that is functionally very different than that of model 1 or model 2. Not all microstates are equally probable, but nor are there any microstates in the full network that are completely irrelevant. Instead, our model gives rise to an "alignment" of sorts, where the most degenerate microstates are also the ones that have the most connected structure in terms of available transitions. This gives rise to the amplified behavior we saw in figure 21. We cannot simply restrict the full Hilbert space of states to these highly degenerate states, but nonetheless they are dominant in determining the dynamics of the system.

For large total winding number $N$, the average total number of strings scales as $N_s \sim \sqrt{N}$ [1, 34]. By contrast, from the degeneracy distribution (red lines) in figure 21, it appears for $N = 20$ we have $N_s \sim 12$. This is actually a remnant of having "small" $N$; at larger $N$ the average $N_s$ value will move towards $\sqrt{N}$.[13]

The bulk dual of the D1-D5 states are the Lunin-Mathur supertubes [13], which (from a six-dimensional perspective) have long but finite AdS$_3$ throats; the length of the throat is determined by the winding numbers of the component strings in the D1-D5 system. In particular, the length of the throat becomes very large when there are mostly strings with large winding number [1]. As discussed above, for large $N$ we expect that the entropically favored geometries have $N_s \sim \sqrt{N}$ (with strings of (large) winding number $\sqrt{N}$), and our network results therefore imply that the typical black hole microstates are simply the entropically-favored geometries that have long throats and look like black holes up until very close to the horizon scale.

Another important feature of our results is that the structure of the network was important for understanding late-time behavior, but the values of the network edge weights turned out to be irrelevant. Intuitively, this tells us that the details of which D1-D5 system states can interact with one another are important, but the actual details of the relative strengths of different interactions are largely irrelevant. This behavior can be understood in the following way. The full D1-D5 gauge theory has a critical point along its RG flow, at which point it is accurately described by an SCFT [1, 33, 34, 55]. Field theories with such conformal critical points are known to exhibit *universality* near these critical points, in the sense that their behavior is determined entirely by the details of the breaking of the conformal symmetry [56, 57]. The string gas states we consider are BPS states at the conformal fixed point, but with some small amount of non-extremality added in order to add dynamics. Intuitively, then, we are probing the theory slightly away from the critical fixed point with relevant deformations. The universal behavior we see is consistent with this intuitive picture, and gives us an indication that our model is a good description of the D1-D5 system in this nearly-conformal regime.

---

[13]This can be confirmed explicitly using the canonical ensemble approximation of sec. 3.2 in [34]; for $N = 20$ we find $\langle N_s \rangle \approx 12.36$ whereas for large $N$ we reproduce $\langle N_s \rangle \sim \sqrt{N}$.

# 6 Discussion

In this section, we discuss certain aspects of our models and their results. We discuss the possible interpretation of the evolution of microstates in our network as "shedding" angular momentum, and we note the link between the trapped phase of model 2 and "glassy" black hole physics. We also touch on various aspects and/or caveats of our analysis with respect to distinguishability of microstates. We end with some comments on future applications of our network theory techniques.

**Angular momentum.**   Physically, the evolution of the network should be thought of as successive tunneling steps in the evolution of the microstate. These tunneling steps are each associated with a change in the properties of the state, such as its energy, its angular momentum, etc. In particular, (as in model 2) if all of our microstates are microstates where all $N$ centers are on a line, then typically microstates with larger $N$ will have larger angular momentum than those with smaller $N$. So, for the trapped phase of model 2, it seems plausible that the microstates are driven to shed their angular momentum as they evolve over time until they reach a stable point at low angular momentum. This shedding of angular momentum is an irreversible relaxation process that is represented by the asymmetry between initial and final states of the transition rates in model 2. It would be interesting to account explicitly for angular momentum in the details of our models in order to investigate this more thoroughly.

The interpretation of the evolution of model 2 as shedding angular momentum is very reminiscent of the discussion in [29].[14] There, a *classical* instability of microstate geometries found in [58] was interpreted as an entropic transition driving (atypical) microstates with large angular momentum to (typical) microstates with smaller angular momentum (for which supergravity ceases to be a good approximation). It is interesting that the instability and evolution there is classical whereas the tunneling transitions we model here are intrinsically quantum. However, the flexible nature of the network model may imply that a very similar model to model 2 with similar evolutions and phases could be used to approximately describe the classical evolution of microstate geometries under this classical instability. It would be very interesting to pursue this further to understand the relation between the dynamics of model 2 and this classical instability evolution.

**Glassy black holes.**   In the trapped phase of model 2, we found that it was possible for random walks on our microstate network to get stuck in long-lived microstates with very few centers. This is very reminiscent of the viewpoint in previous work on glassy black hole physics [32, 59–61] (see also the "supergoop" or "string glasses" of [60]), where one can view a non-extremal multi-centered black hole microstate as a long-lived metastable state, much like glass is. Moreover, one can even find explicit examples of possible ergodicity breaking [60], which is very remniscent of the phase transition between ergodic and trapped behavior we found. The evolution of our network into these trapped states can be interpreted as the evolution of a microstate into local minima of the Hamiltonian that correspond to long-lived (but not absolutely stable) states. It would be interesting to apply the network techniques we have used here to study the evolution of these multi-particle glassy black hole systems in more detail.

It would also be informative to understand the generality of the glassy trapped phase, as that is not immediately clear from our analysis. Model 2 contains glassy phases for particular regions of its parameter space; model 1 is not complex enough to allow for such a glassy phase; and model 3 is "too connected" to allow for such a phase. Glassy phases in network models can likely be related to the existence of sparsely connected communities (as discussed in section 4.2.1) and could possibly be studied using community detection algorithms on networks (see

---

[14]We thank A. Puhm for bringing this to our attention.

also section 2.3); it would be elucidating to analyze the precise conditions that microstate network models need to satisfy to admit such glassy phases.

**Caveat: *N* is not a semi-classical observable.**    In models 1 and 2, we largely focused on the quantity $N$, the number of centers in a given microstate. An important caveat to mention is that this is not a particularly good semi-classical observable, since the number of centers in a given microstate geometry is certainly not a locally measurable quantity. Strictly speaking, it is a global feature of the spacetime in the same sense that the presence of a horizon is a global feature that no local (or even finite) measurement can determine precisely. As such, our results for the expected value of $N$ in the evolution of black hole microstates do not necessarily translate directly into statements about possible observations of such microstates. Similarly, the number of strings $N_s$ that we studied in model 3 is also not a local observable. Nevertheless, $N$ and $N_s$ are interesting quantities in microstates; the network approach we have used is ideal to study them as we only need to input global features into the network models.

**Distinguishibility of microstates.**    Recently, Raju and Shrivastava have argued [62] that the distinguishibility of individual black hole microstates from the thermal average geometry (i.e. the black hole) is exponentially suppressed and hard to measure. (An important earlier work in a similar vein is [63]. See also [64–67] for other work on distinguishing microstates.) We do not address these arguments here as we do not directly discuss distinguishibility between microstates or their thermal average. However, we would like to emphasize that [62] assumes a statistical ensemble and thus a thermodynamic equilibrium (i.e. one without time evolution). As we have mentioned above, our results should be thought of more as understanding the time evolution of black hole microstates, in particular leaving open the possibility of glassy, non-equilibrium evolution. And, as we mentioned above, $N$ is not a semi-classical observable, so the arguments of [62] do not obviously directly apply to $N$ either.

**Other microstate models.**    In this paper, we have created models based on 5D multicentered Bena-Warner microstates and D1-D5 Lunin-Mathur supertube states. An obvious interesting expansion of the scope of these models would be to create and consider models based on the D1-D5-P "superstrata" microstate geometries of [10–12]. Other interesting, related systems include the LLM bubbling geometries [68] that are thought to be microstates of incipient black holes [69–71], and fuzzy D-brane geometries in the BFSS matrix model [72] that are thought to be microstates of Schwarzschild black holes in various dimensions [73–77]. A first step towards understanding their dynamics would be computing tunneling rates between microstates. Tunneling and meta-stable states in LLM geometries have been discussed in prior works [27,78], but for the other examples mentioned the tunneling rates between any geometries have not yet been studied. It would nonetheless be interesting to investigate these other models further.

Another future direction of research would be to extend our network analysis to account for more general classes of Bena-Warner microstates. In model 2, we constructed a network of multi-centered microstate geometries where all charges are placed on a line; the states in this network model should be thought of as compositions of the total charge $Q$ of the black hole, of which there are $2^{Q-1}$ such compositions. For large $Q$, this implies an entropy of $S_{\text{model 2}} \sim Q$, which is less than the scaling of the three-charge black hole entropy $S_{\text{BH}} \sim Q^{3/2}$. This scaling is also less than the scaling of the total number of Bena-Warner multi-centered microstate geometries, which has been argued to be $S_{\text{Bena-Warner}} \sim Q^{5/4}$ [79]. These differences in entropy scaling are unsurprising, since our model only describes solutions where all centers are on a line instead of distributed over $\mathbb{R}^3$. It would be interesting to see if the late-time behavior

we observed in model 2, especially the phase transition, persists when we look at tunneling transitions between these more general Bena-Warner geometries.

**Larger applicability of our methodology.**  Finally, we wish to reiterate that the network theory methods we used in this paper are very general and can be applied to a wide range of quantum mechanical systems. The networks we considered in this paper were constructed to model tunneling between smooth, multi-centered black hole microstates or between different states in the D1-D5 system. However, as shown in section 2.1, there is a very precise way in which networks can be used to model the evolution of generic quantum systems with connected states. Moreover, the main observable we looked at in our models was the late-time probability distribution, but we could easily broaden our scope and study other observables. As discussed in section 2.3, there is a rich literature of network theory tools that can be used to probe other dynamical features of these systems. We are optimistic that the methods presented in this work can be used to better understand tunneling dynamics in other areas of theoretical high energy physics and string theory. We are so far only aware of one such application in string theory, where networks are used to study dynamics in the string landscape [42], but we look forward to seeing more novel applications in the future.

# Acknowledgments

We thank Iosif Bena, Zachary Charles, Emil Martinec, Ruben Monten, Andrea Puhm, and Bert Vercnocke for useful discussions; we are particularly grateful to Andrea Puhm and Bert Vercnocke for detailed comments on an early draft version of this paper. We also especially wish to thank John K. Golden for collaboration in the early stages of this project. We thank Xiao-yue Sun and his collaborators for remarks on v1 of this paper. This work was supported by the U.S. Department of Energy under grant DE-SC0007859. DRM is supported by the ERC Starting Grant 679278 Emergent-BH. AMC is supported in part by the KU Leuven C1 grant ZKD1118 C16/16/005, the National Science Foundation of Belgium (FWO) grant G.001.12 Odysseus, and by the European Research Council grant no. ERC-2013-CoG 616732 HoloQosmos.

# A   Random Walks and Convergence

In this section, we will explain the dynamical random walk method used in sections 4 and 5 to determine when the random walk has converged to a suitably steady-state that approximates the analytic eigenvector centrality well.

## A.1   Theoretical Bounds on Random Walk Convergence

Suppose that our system is initially in the state $\mathbf{p}(0)$, the probability vector whose components $p_i(0)$ are the probability to be at node $i$ at the start of the random walk. The transfer matrix $\mathbf{T}$ of the network generates stochastic time evolution of this state such that

$$\mathbf{p}(t) = \mathbf{p}(0)\mathbf{T}^t. \tag{24}$$

That is, the components $p_i(t)$ of this probability vector are the probability that a random walker is at node $i$ after $t$ discrete time steps. We are interested in the late-time behavior of our network, though, and so we want the eigenvector centrality $\mathbf{p}_\infty$ that is a fixed point of the time evolution such that $\mathbf{p}_\infty = \mathbf{p}_\infty \mathbf{T}$. This is often difficult to compute analytically, though, as it may require explictly inverting the transfer matrix, which can be unfeasible for

networks with a large number of nodes. Instead, we want to approximate the analytic result by performing finite-time random walks. This requires a precise understanding of the *relaxation time* of the network, or the time it takes for $\mathbf{p}(t)$ to closely approximate $\mathbf{p}_\infty$.

Let $\{\lambda_i\}$ be the eigenvalues of the transfer matrix, ordered from largest to smallest magnitude such that $|\lambda_1| \geq |\lambda_2| \geq \ldots \geq |\lambda_N|$. The corresponding orthonormal left eigenvectors $\{\mathbf{v}_i\}$ of the transfer matrix form a complete basis, so we can decompose the initial state of the random walk as

$$\mathbf{p}(0) = \sum_{i=1}^{N} c_i \mathbf{v}_i, \tag{25}$$

for some constants $c_i$. The probability vector after $t$ discrete time steps is then given by

$$\mathbf{p}(t) = \sum_{i=1}^{N} c_i \mathbf{v}_i \mathbf{T}^t = \sum_{i=1}^{N} c_i \lambda_i^t \mathbf{v}_i. \tag{26}$$

The transfer matrix is a stochastic matrix, and so we are guaranteed that all of its eigenvalues have magnitude $|\lambda_i| \leq 1$ and that the largest eigenvalue is $\lambda_1 = 1$. This means that $\mathbf{v}_1$ is the eigenvector centrality, and so

$$\mathbf{p}(t) = c_1 \mathbf{p}_\infty + \sum_{i=2}^{N} c_i \lambda_i^t \mathbf{v}_i. \tag{27}$$

As long as $|\lambda_2| < 1$, the contribution to this from the eigenvectors $\mathbf{v}_i$ for $i > 1$ will become suppressed by successive powers of $\lambda_i$. In the limit where $t \to \infty$, we can ignore these other contributions entirely, and we are left with

$$\lim_{t \to \infty} \mathbf{p}(t) = \mathbf{p}_\infty. \tag{28}$$

The rate of convergence is determined by the magnitude of the eigenvalues of the transfer matrix. If they have small magnitude, then their contribution to the random walk state becomes suppressed very quickly. If they have magnitude close to one, though, then this convergence happens very slowly. It is therefore the parameter $|\lambda_2|$ that controls the convergence rate of this random walk. Convergence, roughly speaking, is achieved after a number of steps $t$ such that

$$|\lambda_2|^t \ll 1. \tag{29}$$

The closer $|\lambda_2|$ is to one, the longer it takes for the random walk to approximate the analytic eigenvector centrality result. The difference $1 - |\lambda_2|$ is referred to as the *spectral gap* of the network; a large spectral gap ensures a fast relaxation time for the network.

The spectral gap is intimately related to the community substructure of the network. If the network contains multiple highly-connected subnetworks that are minimally connected to one another, the transfer matrix would become approximately block diagonal, with each block corresponding to transition rates between nodes of one of the subnetworks. The transfer matrix would then have multiple eigenvalues with $|\lambda| \approx 1$, each corresponding to a steady-state configuration that lives on only one of the subnetworks. Since the subnetworks are minimally connected to one another, it is very unlikely at each time step for a random walker to hop between the subnetworks, and thus the random walk ceases to be a good simulation of the global structure of the network. Intuitively, then, we expect the spectral gap to be smaller for networks with more of these disjoint communities.

Unfortunately, actually computing eigenvalues for most networks generically requires inverting the transfer matrix and is therefore just as difficult as analytically computing the eigenvector centrality. In some cases, there are power methods that can be used to approximate

these eigenvalues that are more efficient than inverting the entire matrix, but even these computations are unfeasible for the networks in model 2 and model 3, since the number of nodes of these networks is exponentially large. Instead, we can set bounds on the magnitude of $\lambda_2$ using results from spectral theory. One such tool for bounding this magnitude is the Cheeger constant $h$ of the network [80]. For a balanced, directed network, this is given by

$$h = \min_S h(S), \quad h(S) = \left[ \frac{\sum\limits_{i \in S, j \in \bar{S}} A_{ij}}{\min\left( \sum\limits_{i \in S} d_i, \sum\limits_{i \in \bar{S}} d_i \right)} \right], \tag{30}$$

where $S$ is a subnetwork, $\bar{S}$ is its complement, and the Cheeger constant is minimized over all choices for $S$. Intuitively, the subnetwork $S$ that minimizes $h(S)$ will be one such that $S$ and $\bar{S}$ are as sparsely connected as possible (minimizing the numerator of (30)) while being roughly even in size (maximizing the denominator). The Cheeger constant is related to the spectral gap of the theory by the Cheeger inequality [81]:

$$\frac{h^2}{2} < 1 - |\lambda_2| \leq 2h. \tag{31}$$

For networks that allow for very sparsely-connected bipartitions, $h$ will be small, so $|\lambda_2|$ will be close to one, and the random walk will take a long time to converge. The Cheeger constant is much simpler to compute than $\lambda_2$, since it doesn't require inverting large matrices, and since we can obtain an estimate for $h$ just by using particular nice choices of $S$.

To see this in action, let's first consider model 1. For the range of parameters where $\beta \leq 0$, $\delta \leq 0$, and $\gamma \geq 0$, the Cheeger constant corresponds to $h(S)$ for the subnetwork $S = \{1, \ldots, N_{\max}/2\}$. Importantly, on the relevant subset of parameter space $\gamma = 1$, $-1 \leq \beta \leq 0$, and $-2 \leq \delta \leq 0$, we find the following numerical bounds on $h$:

$$\begin{aligned}
N_{\max} = 10 : &\quad h \geq 0.071, \\
N_{\max} = 20 : &\quad h \geq 0.034, \\
N_{\max} = 50 : &\quad h \geq 0.013.
\end{aligned} \tag{32}$$

That is, $h$ is always positive on our parameter space, so the Cheeger inequality tells us that $1 - |\lambda_2| > 0$, and thus the random walk on this network is guaranteed to converge eventually. Additionally, the Cheeger inequality can be used to estimate how quickly the random walk converges, because it gives us the relation

$$(1 - 2h)^t \leq |\lambda_2|^t < \left( 1 - \frac{h^2}{2} \right)^t. \tag{33}$$

This expression gives a lower-bound and an upper-bound on the number of steps required to ensure that $|\lambda_2|^t \ll 1$. For example, in the case of $N_{\max} = 10$, if we want $|\lambda_2|^t \approx 0.01$ as our convergence condition, we can plug (32) into (33) and compute that we would need somewhere between $t \approx 30$ and $t \approx 1840$ steps in order for the random walk to converge.

We can do a similar analysis for model 2. For simplicity, we will look at the simple case where the degeneracies $\omega$ and the transition rates $\Gamma$ are all set to one. For networks with $Q \leq 20$, we find that the Cheeger constant corresponds to the subnetwork $S$ that contains all nodes with $N \leq Q/2 - 1$. This matches our intuitive notion that $h(S)$ is minimized for a bipartition that splits the network into roughly two equal-size parts. The corresponding

Cheeger constants for some values of $Q$ are given below:

$$\begin{aligned}
Q = 10: \quad & h = 0.24, \\
Q = 12: \quad & h = 0.21, \\
Q = 14: \quad & h = 0.18, \\
Q = 16: \quad & h = 0.17, \\
Q = 18: \quad & h = 0.16, \\
Q = 20: \quad & h = 0.15.
\end{aligned} \tag{34}$$

These are all not too small, indicating that convergence is relatively easy. As expected, convergence becomes harder for larger $Q$ simply because of the exponential growth of microstates with $Q$. Even for the $Q = 20$ network, though, we can use (33) to estimate that after $t = 10,000$ steps, the finite-time probability approximates the late-time probability with an error of $|\lambda_2|^t \leq 7 \times 10^{-50}$. Of course, this analysis was done in the special case where $\omega = \Gamma = 1$ for simplicity. However, the results are not changed drastically by introducing parameter dependence. We can therefore trust that the random walks that we do in section 4 (see e.g. figure 15) are good approximations of the late-time behavior of model 2, since they are typically run for more than $t = 10,000$ steps.

We can also compute the Cheeger constant for model 3, although due to its similarities to model 2, the results are very similar. The upshot is that all of our models have good theoretical convergence rates, and and thus random walks (as long as they run for a sufficiently long time) will eventually converge to the analytic centrality.

## A.2 Dynamic Random Walk Method

Now that we have established that the random walk probability $\mathbf{p}(t)$ on the networks considered in this paper will converge to the analytic late-time result $\mathbf{p}_\infty$ after a sufficiently large (but finite) number of steps, we need to determine how many steps to actually make the random walk go through. Instead of setting a hard cut-off on the number of steps, we instead use a dynamic random walk method that tests for convergence as the random walk progresses and stops when it has converged within some small error threshold. The details of this dynamical method are as follows.

Let $f_i(t)$ be the fraction of steps spent in node $i$ for a random walk that has run for $t$ discrete time steps. If $f_i(t+1)$ is significantly different from $f_i(t)$, we cannot expect the random walk results to be stable, so we cannot expect $f_i(t)$ to be close to the true probability $p_i(t)$. If the random walk has run for a large number of steps and this fraction is stable, then it will serve as a good approximation. Moreover, since $p_i(t)$ approaches $p_{\infty,i}$ at late times, we can conclude that $f_i(t) \approx p_{\infty,i}$ if enough time steps have been run for enough steps such that $|\lambda_2|^t \ll 1$ and the random walk is stable.

In models 2 and 3, we are primarily interested in the probability that the system is in a group of microstates with a particular value of $N$, where $N$ is the number of centers in model 2 or the number of strings in model 3. So, we do not actually have to have strict convergence for each individual microstate fraction $f_i(t)$. Instead, we will define

$$\begin{aligned}
f(N, t) &= \sum_{i \mid n(i) = N} f_i(t), \\
p_\infty(N) &= \sum_{i \mid n(i) = N} p_{\infty,i},
\end{aligned} \tag{35}$$

as the random walk fraction and analytic probabilities, respectively, for the system to be in any state $i$ with $n(i) = N$ centers or strings, respectively. We will therefore only have to test

for a more limited version of convergence, where $f(N, t)$ is not significantly different from $f(N, t + 1)$, in which case we conclude that the random walk serves as a good approximation for the analytic result, i.e. $f(N, t) \approx p_\infty(N)$. Our results will look similar whether we test for convergence in $N$ or in each individual node, but convergence in $N$ is much faster to achieve than convergence for each individual node.

To concretely test for this convergence, we will set a number of bins $n_b$ to put the data into such that each bin contains $\Delta t = \frac{t}{n_b}$ data points. The coarse-grained average of the random walk fraction $f(N, t)$ within each bin is then given by

$$\langle f(N, t) \rangle_{\text{bin } n} = \frac{1}{\Delta t} \sum_{t=(n-1)\Delta t}^{n\Delta t} f(N, t). \tag{36}$$

This coarse-grained fraction is a better way to study convergence, because all random walks will inherently have some small fluctuations over time as they rarely stay at the same node at any consecutive time steps. A random walk will have converged when all of these coarse-grained fractions are comparable. Of course, we should not necessarily include the first few bins, since those are the ones that are sensitive to the initial conditions of the random walk. Instead, we will mark bin $n_c$ as the one we start comparing from, and then we look at the last $n_b - n_c + 1$ bins. We will define the absolute error $E_{\text{abs}}(N, t)$ and the relative error vector $E_{\text{rel}}(N, t)$ of these bins to be as follows:

$$
\begin{aligned}
E_{\text{abs}}(N, t) &= \max_{n=n_c,\dots,n_b} \left| \langle f(N, t) \rangle_{\text{bin } n_c} - \langle f(N, t) \rangle_{\text{bin } n} \right|, \\
E_{\text{rel}}(N, t) &= \max_{n=n_c,\dots,n_b} \left| \frac{\langle f(N, t) \rangle_{\text{bin } n_c} - \langle f(N, t) \rangle_{\text{bin } n}}{\langle f(N, t) \rangle_{\text{bin } n_c}} \right|.
\end{aligned}
\tag{37}
$$

That is, the absolute error computes the maximum absolute change in $\langle f(N, t) \rangle$ among these bins, while the relative error computes the maximum fractional difference of $\langle f(N, t) \rangle$ among these bins[15]. These quantities serve as an estimate for how much the random walk results could be affected by running it for another $\Delta t$ steps. The random walk is deemed to have converged to a particular threshold when we find that

$$E_{\text{abs}}(N, t) \le \Sigma_{\text{abs}} \quad \text{and} \quad E_{\text{rel}}(N, t) \le \Sigma_{\text{rel}}, \tag{38}$$

for all allowed values of $N$. That is, the coarse-grained absolute and relative errors associated with $f(N, t)$ have to be below a specified threshold for all values of $N$ simultaneously before the random walk is ended.

In sections 4 and 5, we tested our random walks for convergence by coarse-graining the random walk fractions into $n_b = 20$ bins. We also set $n_c = 11$ in order to compare the coarse-grained averages in the last ten bins. Additionally, the error thresholds we used were $\Sigma_{\text{abs}} = \Sigma_{\text{err}} = 10^{-4}$. This means that if our random walk converged after $t = 20,000$ steps, we would observe a change of at most one part in $10^{-4}$ difference in our results if we ran the random walk for another $\Delta t = 1000$ steps. Tightening the error threshold beyond these values did not end up changing our results significantly in any way. Moreover, increasing the number of coarse-grained bins made the random walk take longer to converge, but with the same results. The error parameters we used led to efficient, accurate results that converged well after the relaxation time of the network, which leads us to conclude that these choices are reasonable to use.

---

[15] Note that we want to understand both types of errors, because the absolute and relative errors become more important for larger and smaller values of $f(N, t)$, respectively.

# B   Explicit Black Hole Microstate Calculations

In this appendix, we present some explicit calculations pertaining to black hole microstate geometries that are relevant for the setup of the network models (as presented in section 1.1). We first review the construction of the Bena-Warner five-dimensional microstate geometries in supergravity, including the bubble equations, and then we go on to show an explicit construction of microstate splitting. We end with a numerical estimation of tunneling rate parameters in model 2 using properties of a number of explicit microstate geometries.

## B.1   Supergravity Setup and Multi-center Solutions

The solutions we will consider are supersymmetric solutions in 5D minimal supergravity coupled to two vector multiplets. It contains vectors $A^I$ (with corresponding field strengths $F^I$) and scalars $y^I$ (using the conventions of [2]):

$$S_5 = \frac{1}{16\pi G_5} \int \left( \star_5 R_5 - Q_{IJ} \star_5 dy^I \wedge dy^J - Q_{IJ} \star_5 F^I \wedge F^J - \frac{1}{6} C_{IJK} F^I \wedge F^J \wedge A^K \right), \quad (39)$$

where $I = 1, 2, 3$. The vector multiplet kinetic matrix is

$$Q_{IJ} = \frac{1}{2}(y^I)^{-2}\delta_{IJ}. \quad (40)$$

The scalars obey the restriction

$$\frac{1}{6}C_{IJK}y^I y^J y^K = 1, \quad (41)$$

with

$$C_{IJK} = |\epsilon_{IJK}|. \quad (42)$$

The metric, scalars and gauge fields of supersymmetric solutions with a timelike Killing vector have the form [82,83]:

$$ds_5^2 = -(Z_1 Z_2 Z_3)^{-2/3}(dt + k)^2 + (Z_1 Z_2 Z_3)^{1/3} ds_4^2, \quad (43)$$

$$y^I = \frac{(Z_1 Z_2 Z_3)^{1/3}}{Z_I}, \quad (44)$$

$$A^I = \left(-Z_I^{-1}(dt + k) + B^I\right). \quad (45)$$

The forms $k$ and $B^{(I)}$ and the warp factors $Z_I$ are supported on and only depend on the 4D Gibbons-Hawking (GH) base space, which has metric:

$$ds_4^2 = V^{-1}(d\psi + A)^2 + V ds_3^2(\mathbb{R}^3), \qquad \star_3 dA = -dV, \quad (46)$$

with $V$ a harmonic function on $\mathbb{R}^3$.[16] The solution is completely determined by 8 harmonic functions $(V, K_I, L^I, M)$ on $\mathbb{R}^3$ [84,85] which enter the fields as:

$$
\begin{aligned}
B^I &= V^{-1}K^I(d\psi + A) + \xi^I, & d\xi^I &= -\star_3 dK^I \\
Z_I &= L_I + \frac{1}{2}D_{IJK}V^{-1}K^J K^K \\
k &= \mu(d\psi + A) + \omega, & \mu &= \frac{1}{6}V^{-2}C_{IJK}K^I K^J K^K + \frac{1}{2}V^{-1}K^I L_I + M,
\end{aligned}
$$

$$\star_3 d\omega = V dM - M dV + \frac{1}{2}(K^I dL_I - L_I dK^I). \quad (47)$$

---

[16]5D solutions with a GH base have a natural interpretation upon KK reduction along the GH fibre $\psi$ as 4D multi-center solutions.

If the harmonic functions have sources at $N$ centers at coordinates $\vec{r}_i$ in $\mathbb{R}^3$, then we have:

$$V = \sum_{i=1}^{N} \frac{v_i}{|\vec{r} - \vec{r}_i|}, \qquad\qquad M = m_0 + \sum_{i=1}^{N} \frac{m_{0,i}}{|\vec{r} - \vec{r}_i|}, \qquad (48)$$

$$K^I = \sum_{i=1}^{N} \frac{k_i^I}{|\vec{r} - \vec{r}_i|}, \qquad\qquad L_I = 1 + \sum_{i=1}^{N} \frac{\ell_{I,i}}{|\vec{r} - \vec{r}_i|}. \qquad (49)$$

The only free parameters are the KK monopole charges $v_i$ and dipole charges $k_i^I$ ; smoothness of the solutions at the different centers $\vec{r}_i$ fixes the sources of $L_I$ and $M$:

$$\ell_{I,i} = -\tfrac{1}{2} C_{IJK} \frac{k_i^J k_i^K}{v_i}, \qquad m_i = \frac{1}{2} \frac{k_i^1 k_i^2 k_i^3}{q_i^2} \qquad \forall i \ (\text{no sum}). \qquad (50)$$

Five-dimensional Minkowski asymptotics requires $\sum_{i=1}^{N} v_i = 1$ and fixes the constants of the harmonic functions by $V|_\infty = K^I|_\infty = 0$, $L_I|_\infty = 1$ and $M|_\infty = m_0$ with

$$m_0 = -\frac{1}{2} \sum_{I=1}^{3} \sum_{i=1}^{N} k_i^I. \qquad (51)$$

The charges $v_i, k_i^I$ and the positions of the centers $\vec{r}_i$ cannot be chosen arbitrarily: to ensure that the solution does not have closed timelike curves (CTCs), the so-called *bubble equations* must be satisfied for each center $i$ [2]:

$$\sum_{j \neq i} \left( \frac{k_j^1}{v_j} - \frac{k_i^1}{v_i} \right) \left( \frac{k_j^2}{v_j} - \frac{k_i^2}{v_i} \right) \left( \frac{k_j^3}{v_j} - \frac{k_i^3}{v_i} \right) \frac{v_i v_j}{r_{ij}} = -2 \left( m_0 v_i + \frac{1}{2} \sum_{I=1}^{3} k_i^I \right), \qquad (52)$$

where $r_{ij} \equiv |\vec{r}_i - \vec{r}_j|$ is the distance between centers $i$ and $j$. The bubble equations give $N-1$ independent constraints on the variables $v^i, k_i^I, \vec{r}_i$; the sum of the bubble equations vanishes when (51) holds.

The physical, asymptotic charges are normalized as

$$Q_I \equiv \frac{1}{4\pi^2} \int Q_{IJ} \star_5 F^J = -2 C_{IJK} \sum_j \frac{\tilde{k}_j^J \tilde{k}_j^K}{v_j}, \qquad \tilde{k}_j^I \equiv k_j^I - v_j \sum_k k_k^I, \qquad (53)$$

which gives asymptotically $Z_I = Q_I / \rho^2$ for the radius $\rho$ in standard polar coordinates on a constant time slice at infinity.

In 5D, there are also two angular momenta. A convenient parametrization for these is given by $J_R, J_L$ with:

$$J_R = \frac{4}{3} C_{IJK} \sum_i \frac{\tilde{k}_i^I \tilde{k}_i^J \tilde{k}_i^K}{v_i^2}. \qquad (54)$$

The expression for $J_L$ is a bit more involved. In the special case where all centers are on a line and ordered from 1 to $N$, the magnitude of $J_L$ is given by:

$$J_L = \frac{4}{3} C_{IJK} \sum_{1 \leq i \leq j \leq N} v_i v_j \left( \frac{k_j^I}{v_j} - \frac{k_i^I}{v_i} \right) \left( \frac{k_j^J}{v_j} - \frac{k_i^J}{v_i} \right) \left( \frac{k_j^K}{v_j} - \frac{k_i^K}{v_i} \right). \qquad (55)$$

## B.2 Explicit Splitting of Multi-centered Microstates

In this section, we want to construct an explicit example of a $N$-center solution where one (or a few) centers "split" into multiple centers, creating a $N' > N$ center solution, and where all asymptotic charges are kept fixed. This will serve as proof of principle that such a splitting of centers in a multicentered solution is at the very least a physical possibility and does not necessarily require changing the asymptotic charges. Note that [27] considers the formation of an $N'$-centered solution by considering intermediate $N$-center solutions and the transitions between them. However, the transitions considered there do not necessarily keep the asymptotic charges fixed; in fact, the angular momentum for the two species of solutions considered in [27] differs for varying $N$. Along the way, we will also find expressions which quantify how much it is possible to separate the contributions of different centers (or small collections of centers) to the total asymptotic charges of the solution.

Let us first develop a general framework that is useful to consider when "splitting" microstates of different numbers of centers. We will generically consider a multi-centered solution that consists of $N + 2m$ centers. The $N$ centers with $i = 1, \ldots, N$ are considered to be a "black hole blob" [86] with:

$$\sum_{i=1}^{N} v_i = +1, \tag{56}$$

and we define:

$$\hat{Q}_I = -2C_{IJK} \sum_{i=1}^{N} \frac{\hat{k}_i^J \hat{k}_i^K}{v_i}, \qquad \hat{k}_i^I = k_i^I - v_i \hat{k}_0^I, \qquad k_0^I = \sum_{i=1}^{N} k_i^I, \tag{57}$$

where $\hat{k}_i^I$ and thus $\hat{Q}_I$ is gauge invariant (due to (56)). $\hat{Q}_I$ can be interpreted as the charge that the $N$ center "blob" contributes to the total charge $Q_I$ in (53) of the complete $N + 2m$ center solution. The angular momenta of the blob are:

$$\hat{J}_R = \frac{4}{3} C_{IJK} \sum_{i=1}^{N} \frac{\hat{k}_i^I \hat{k}_i^J \hat{k}_i^K}{v_i^2}, \tag{58}$$

and if the centers are all on a line:

$$\hat{J}_L = \frac{4}{3} C_{IJK} \sum_{1 \le i \le j \le N} v_i v_j \left( \frac{k_j^I}{v_j} - \frac{k_i^I}{v_i} \right) \left( \frac{k_j^J}{v_j} - \frac{k_i^J}{v_i} \right) \left( \frac{k_j^K}{v_j} - \frac{k_i^K}{v_i} \right). \tag{59}$$

The $2m$ other centers are divided into $m$ "supertube pairs". The $k$-th supertube consists of the centers numbered $c_1^{(k)} \equiv N + 2(k-1) + 1$ and $c_2^{(k)} = N + 2(k-1) + 2$ and has GH charges:

$$v_{c_1^{(k)}} = -Q_k, \qquad v_{c_2^{(k)}} = +Q_k. \tag{60}$$

It will be useful to define the (gauge-invariant) quantities $d_k^I, f_k^I$ for each supertube:

$$d_k^I \equiv 2(k_{c_1^{(k)}}^I + k_{c_2^{(k)}}^I), \qquad f_k^I \equiv 2k_0^I + \left( 1 + \frac{1}{Q_k} \right) k_{c_1^{(k)}}^I + \left( 1 - \frac{1}{Q_k} \right) k_{c_2^{(k)}}^I. \tag{61}$$

We also define:

$$j_{R,k} \equiv \frac{1}{2} C_{IJK} (f_k^I f_k^J d_k^K + f_k^I d_k^J d_k^K) - \frac{1}{24} (1 - Q_k^{-2}) C_{IJK} d_k^I d_k^J d_k^K, \tag{62}$$

$$j_{L,k} \equiv \frac{1}{2} C_{IJK} (d_k^I f_k^J f_k^K - f_k^I d_k^J d_k^K) + \left( \frac{3Q_k^2 - 4Q_k + 1}{24Q_k^2} \right) C_{IJK} d_k^I d_k^J d_k^K. \tag{63}$$

If there is only one supertube, $m = 1$, the charges $Q_I, J_R, J_L$ (where the latter is only valid if the centers are on a line, with the supertube to the right of the blob) can be written as:

$$Q_I = \hat{Q}_I + C_{IJK} d^J f^K, \tag{64}$$

$$J_R = \hat{J}_R + d^I \hat{Q}_I + j_R, \tag{65}$$

$$J_L = \hat{J}_L - d^I \hat{Q}_I + j_L. \tag{66}$$

These expressions were first given in [86]. To consider the case of two supertubes, $m = 2$, one way to calculate the total charge is to realize that we can consider the $N + 2$ centers consisting of the black hole blob plus the first supertube as an "effective blob" (since adding the supertube does not spoil (56)) and use (64); then, we can split off the supertube from the effective blob using (64) once again. This gives a straightforward way to generalize (64) to $m = 2$, which in turn can be used to generalize (64) to $m = 3$, and so on.

For $m = 2$, the charges $Q_I, J_R$ are given by:[17]

$$Q_I = \hat{Q}_I + C_{IJK} d_1^J f_1^K + C_{IJK} d_2^J f_2^K + C_{IJK} d_1^J d_2^K, \tag{67}$$

$$J_R = \hat{J}_R + (d_1^I + d_2^I)\hat{Q}_I + j_{R,1} + j_{R,2} + \frac{1}{2} C_{IJK} d_1^I d_2^J (2f_1^K + 2f_2^K + d_1^K + d_2^K). \tag{68}$$

For example, for the charges $Q_I$, with respect to the one-supertube case (64), we notice that simply adding up the two extra contributions ($\sim C_{IJK} d^J f^K$) of the two individual supertubes is not enough; there is also a *cross-term* $C_{IJK} d_1^J d_2^K$ present in the expression for the charge, which comes from the charge generated by the dipole-dipole interaction between the two supertubes.

We also give the expressions for $Q_I, J_R$ for $m = 4$ where the 3rd, resp. 4th supertube has identical charges to the 1st, resp. 2nd supertube:

$$Q_I = \hat{Q}_I + 2C_{IJK}(d_1^J f_1^K + d_2^J f_2^K) + C_{IJK}(d_1^J d_1^K + d_2^J d_2^K) + 4C_{IJK} d_1^J d_2^K, \tag{69}$$

$$J_R = \hat{J}_R + 2(d_1^I + d_2^I)\hat{Q}_I + 2(j_{R,1} + j_{R,2}) + C_{IJK} d_1^I d_1^J (d_1^K + 2f_1^K)$$
$$+ C_{IJK} d_2^I d_2^J (d_2^K + 2f_2^K) + C_{IJK} d_1^I d_2^J (4f_1^K + 4f_2^K + 6d_1^K + 6d_2^K). \tag{70}$$

Now there are cross-terms between different kinds of supertubes as well as between the pairs of identical supertubes; care must be taken to identify the correct combinatorial factor that these cross-terms appear with. The $m = 2$ and $m = 4$ expressions given above in (64)-(66) and (67)-(68) are generalizations of the $m = 1$ expressions of [86] and have not yet appeared elsewhere in the literature to the best of our knowledge.

Now, to give a proof of principle that it is possible to "split" a multi-centered solution from $N$ centers to a solution with $N' > N$ centers with exactly the same asymptotic charges, we will consider an explicit case of a solution with 7 centers (a blob of 3 centers and 2 identical supertubes) to 11 centers (a blob of 3 centers and 2 *pairs* of identical supertubes). Both the 7- and 11-center configurations have all centers on the $z$-axis and are $\mathbb{Z}_2$ symmetric around $z = 0$, which implies $J_L = 0$. We have checked that both solutions satisfy the bubble equations[18] and are manifestly free of CTCs everywhere.

**7 center solution:**

| # | 1 | 2 | 3 | 4 | 5 | 6 | 7 |
|---|---|---|---|---|---|---|---|
| $v_i$ | +1 | −1 | +1 | −1 | +1 | −1 | +1 |
| $k_i^I$ | −60 | 80 | −60 | 80 | −60 | 80 | −60 |
| $z_i$ | −274.96 | −215.52 | −88.00 | 0 | 88.00 | 215.52 | 274.96 |

---

[17]It is also straightforward to obtain expressions for $J_L$ if the centers are all on a line (also in the $m = 4$ case), but we will not need those expressions here.

[18]In the 7- and 11-center solutions below, all quantities are given only to within a given precision; the bubble equations were solved to a much greater precision than given.

The charges associated to this solution are:

$$Q_I = 19200, \qquad J_R = 5.376 \times 10^6. \tag{71}$$

The pairs of centers (2,1) and (6,7) are the identical supertubes which we take to be the initial $m = 2$ supertubes; each of these will split into two new supertubes. The "black hole blob" is given by the middle three centers (3,4,5). The relevant parameters of the initial supertubes and the black hole blob are:

$$d_0^I = 40, \qquad f_0^I = 80, \qquad k_0^I = -40, \qquad \hat{Q}_I = 3200, \qquad \hat{J}_R = 3.84 \times 10^5. \tag{72}$$

**11 center solution:**

| # | 1 | 2 | 3 | 4 | 5 | 6 | 7 | 8 | 9 | 10 | 11 |
|---|---|---|---|---|---|---|---|---|---|---|---|
| $v_i$ | +1 | −1 | +1 | −1 | +1 | −1 | +1 | −1 | +1 | −1 | +1 |
| $k_i^I$ | −72.38 | 79 | −58.80 | 72.5 | −60 | 80 | −60 | 72.5 | −58.80 | 79 | −72.38 |
| $z_i$ | −1050.61 | −1049.23 | −166.04 | −140.71 | −120.90 | 0 | 120.90 | 140.71 | 166.04 | 1049.23 | 1050.61 |

This solution has exactly the same $Q_I, J_R$ (and $\hat{Q}, \hat{J}_R$, $k_0$ associated to the middle "blob" of centers (5,6,7)) as the 7-center one above, per construction. Each of the two (identical) supertubes from the 7-center solution have split into two identical pairs of two supertubes $\{(8,9),(10,11)\}$ and $\{(4,3),(2,1)\}$. The two supertubes (8,9) and (4,3) have parameters $d_1, f_1$ and the two supertubes (10,11) and (2,1) have parameters $d_2, f_2$, given by:

$$d_1 = 27.3902, \qquad\qquad f_1 = 65 \tag{73}$$
$$d_2 = 13.2322, \qquad\qquad f_2 = 78. \tag{74}$$

### B.3 Estimating parameters in Model 2

In this section, we will get a rough estimate of the tunneling rate parameters $\lambda$ and $\omega$ in (17) of model 2 by fitting those parameters to the tunneling rate in a particular class of multi-centered solutions. The solutions we will consider have $N$ centers on a line where each GH center has alternating charge $\pm 1$:

$$v_i = (-1)^{i-1}, \tag{75}$$

and we choose $N$ odd so that $\sum_i v_i = +1$. The three $k_i^I$ charges for each center are all equal and given by:

$$k_i^I = -v_i N \hat{k} + \hat{k}, \tag{76}$$

so that $\sum_i k_i^I = 0$ and the physical flux between two centers is

$$\Pi_{ij}^I = (v_j - v_i)\hat{k}. \tag{77}$$

The asymptotic charges in this background are all equal and are given by:

$$Q_I = -4 \sum_j v_j (-v_j N \hat{k} + \hat{k})^2 = 4\hat{k}^2(N^2 - 1), \tag{78}$$

so $\hat{k}$ is a parameter that is determined if we are given a fixed $Q_I, N$. A graphical representation of this solution is given in fig. 22. We will only consider $N = 4k + 3$ for some integer $k$, so that we can put the middle center at the origin and retrieve a $\mathbb{Z}_2$-symmetric solution. The positions of all other centers are then completely determined by the bubble equations.

The tunneling probability to tunnel a supertube carrying charge $Q$ off of a center $i$ to the adjacent center $j$ was calculated in [27] to be:

$$\Gamma \sim \exp(-B), \qquad B \sim Q\, r_{ij}, \tag{79}$$

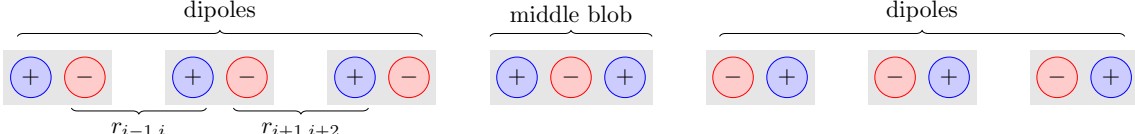

$$r_{i-1,i} \qquad r_{i+1,i+2}$$

Figure 22: The black hole microstate of $N$ centers on a line with $v_i = (-1)^{i-1}$ and $k_i^I = -v_i N \hat{k} + \hat{k}$; positive and negative values of $v$ are drawn in blue and red, respectively. The centers organize themselves into dipoles, except the "middle blob" of three centers. The distance between each pair of centers that makes up a dipole has a tendency to be much smaller than the distance between the dipoles. The distance $r_{d,\text{avg}}(i)$ as defined in (80) is the average of the distances $r_{i-1,i}$ and $r_{i+1,i+2}$

where $r_{ij}$ is the (coordinate) distance (in $\mathbb{R}^3$) between the two centers $i$ and $j$.[19] This immediately implies $\delta = 1$ (since $r_{ij}$ does not depend on the supertube's charge $Q$). The distance $r_{ij}$ is determined by the interaction of the fluxes on the centers through the bubble equations and should therefore be related to the parameters $\lambda$ and $\omega$ in the tunneling rate (17) of model 2.

Let us try to find a rough estimate for these parameters by investigating how $r_{ij}$ changes according to how much charge is to the left and right of it. We will consider solutions with $N = 15, 19, 23, 27, 31, 35, 39, 43, 47, 51, 55, 59, 63, 67$ centers. Within these solutions we will compute $r_{d,\text{avg}}(i)$, the average distance between the dipole made out of centers $i$ and $i+1$ and its neighboring dipoles, as depicted in figure 22. That is,

$$r_{d,\text{avg}}(i) \equiv \frac{1}{2}\left(r_{i-1,i} + r_{i+1,i+2}\right). \tag{80}$$

Since the $i$-th and $(i+1)$-th centers form a dipole together, we are essentially considering the average of the distances between that dipole and the two neighboring dipoles. To make these computations tractable, we will just consider $i = 3, 5, 7$. In terms of the network parameters, this distance should be given by

$$r_{d,\text{avg}}^{(\text{est})}(i) = a\left[Q_L(i)Q_R(i)\right]^{\lambda}, \tag{81}$$

where we define

$$Q_L(i) = \sum_{k=0}^{\frac{i-1}{2}-1} Q_d\,\omega^k, \tag{82}$$

so that $Q_L$ is the sum over all dipole charges $Q_d$ to the *left* of center $i$; each successive dipole has an extra attenuating factor $\omega$. $Q_R$ similarly involves a sum over contributions from all dipoles to the *right* of the dipole $(i, i+1)$, but it also includes the contribution from the middle blob of three centers with charge $Q_{\text{mid}}$, giving:

$$Q_R(i) = \sum_{k=0}^{\text{mid}(i)-1} Q_d\,\omega^k + \omega^{\text{mid}(i)}\left(Q_{\text{mid}} + \sum_{k=1}^{\frac{N-3}{4}} Q_d\,\omega^k\right), \tag{83}$$

$$\text{mid}(i) = \frac{N-3}{4} - \left(\frac{i-1}{2} + 1\right). \tag{84}$$

For the solutions we are considering, we have:

$$Q_d = -4N\hat{k}^2, \qquad Q_{\text{mid}} = (N^2 - 6N + 1)\hat{k}^2. \tag{85}$$

---

[19]In fact, in [27], it was derived that $B \sim |d|\,r_{ij}$ where $d$ is the dipole of the supertube. This is related to the charge carried by the supertube as $d \sim Q\hat{k}^{-1}$. We are keeping $\hat{k}$ fixed which implies (79).

We can now equate $r_{d,\text{avg}}^{(\text{est})}(i)$ of (81) to $r_{d,\text{avg}}(i)$ of (80) by fitting the parameters $a, \lambda, \omega$. In principle, we have $42 = 14 \times 3$ datapoints (14 different values for $N$ and three for $i$). However, for a given $\omega$, we delete any datapoints where $Q_L Q_R < 0$ as such points would not make sense. The best-fit (using a $\log-\log$ fit) gives as parameter values (using $\hat{k} = 10$ and $\tilde{a} \equiv \log(a\hat{k}^{4\lambda})$):

$$\omega \approx 0.37, \qquad \lambda \approx -0.18, \qquad \tilde{a} \approx 5.94. \tag{86}$$

A comparison between the fitted network theory values and the explicit microstate values is depicted in figure 23. For this $\log-\log$ fit, we have $1 - R^2 \approx 0.24$.

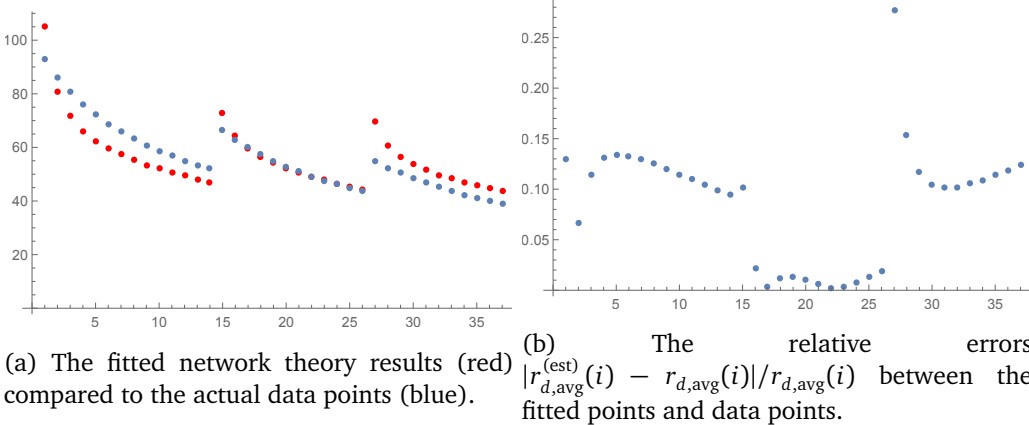

(a) The fitted network theory results (red) compared to the actual data points (blue).

(b) The relative errors $|r_{d,\text{avg}}^{(\text{est})}(i) - r_{d,\text{avg}}(i)|/r_{d,\text{avg}}(i)$ between the fitted points and data points.

Figure 23: Comparison of the data points and the fit

One should tread extremely careful when trying to attach actual physical meaning to these results. The model that we have used to determine $\lambda$ and $\omega$ is very rough, and it is wholly too simple to capture the intricacies of the actual interplay due to the bubble equations between intercenter distances and the charges (as indeed the relatively high value for $1 - R^2$ of the above fit indicates). Moreover, we have only considered a limited, very special class of multi-centered microstate solutions in determining the parameters. Nevertheless, our results seem to indicate that $\lambda \approx -0.18$ is an approximation for the physical interactions of our multi-centered microstates; in particular, $\lambda$ is *negative*. We also have indications that the most physically relevant value of $\omega$ is $\omega \approx 0.37$, which gives an indication of how much the effects of charges on the intercenter distance attenuate as the charge gets further from the centers considered.

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
