# Peer review of "Probing Black Hole Microstate Evolution with Networks and Random Walks"

_SciPost Physics, doi:SciPost Phys. 8, 077 (2020)_

## Round 2 · Referee Report · Anonymous (Referee 1) · 2020-3-11

Report

The article tackles the physically interesting but challenging problem of black hole formation and thermalisation in the framework of the fuzzball conjecture. The objective is to study the process of quantum tunneling between the different microstates of a near-extremal black hole and to understand qualitative and quantitative features of the state reached when the system has attained equilibrium. Since this is a difficult problem to approach with the standard analytic techniques, the authors use the method of network models, which is not very well-known in the string theory community. Each node of the network represents a microstate and each link a possible quantum transition between the microstates. The method uses as inputs the degeneracy of each node and the tunneling amplitudes between the nodes. The authors make some simplifying assumptions to determine these inputs, partly using microscopic and supergravity calculations, and identify three different models, which are based on a different representation of the black hole near-extremal microstates. The first two models are based on the multi-center three-charge microstate geometries, while the last one uses the D1-D5 microsates. The authors show that the different models lead to qualitatively different relaxation behaviours: in some cases the dynamics explores ergodically the whole microstate phase-space, while in others the evolution is trapped in certain subspaces of the whole phase-space.

The article address a very relevant physical problem with an original and well-motivated approach, and it reaches clear and interesting conclusions. The material is very clearly presented, and it contains a nice and self-contained review both of network theory and of the microstate geometry program. I thus recommend the article for publication.

Requested changes

I would like to raise a few questions/comments to the authors.

1) I would like the authors to clarify the relation between eq. (1.1) and the previous statement that the tunneling amplitude should be of order e^{-S}; in particular, how does the entropy should scale with the number N of centers?

2) In model 3, the authors find that in the equilibrium configuration the number N_s of strings is roughly of order N/2. This is much larger than the average number of strings in a typical state of the D1-D5 statistical ensemble, which is of order \sqrt{N}, as explained for example in ref. [1]. Do the authors have any explanation for this apparent discrepancy?

3) I believe that the comment contained in the paragraph starting with "The bulk dual of the D1-D5 states are ..." and ending with "... until very close to the horizon scale." (section 5.3) is not correct. The microstates with the longest throats, and that thus look more like the black hole, are the ones with few strings with large winding numbers, which is the opposite of what is said in section 5.3. See for example eq. 9.176 of ref. [1]. I would like to ask the authors to correct this statement.

  • validity: -
  • significance: -
  • originality: -
  • clarity: top
  • formatting: -
  • grammar: -

Author:  Daniel Mayerson  on 2020-03-28  [id 780]

(in reply to Report 1 on 2020-03-11)
Category:
answer to question
correction

We thank the referee for their positive and constructive feedback and thoughtful comments. To reply to their specific questions/comments:

Regarding 1): Equation (1.1) was essentially a simplified version of equation (1.4) in reference [27]. However, as the referee correctly points out, it was not obvious in the way our equation was written what the relation was between (1.1) and the statement that the tunneling amplitude should be of order e^{-S}. We have altered eq (1.1) (and the immediately ensuing discussion) to better reflect [27], eq (1.4), thus including the black hole entropy explicitly in the exponent of the tunneling amplitude.

Regarding 2) and 3): We thank the referee for bringing this to our attention. In fact, the first and second paragraphs of sec 5.3 contained a few misleading and/or erroneous statements (including that which the referee points out in his point 3). All but the first and last paragraphs of sec 5.3 in the new version have been rewritten to ameliorate this.

The average or expected value of the number of strings N_s in equilibrium follows the degeneracy peak (i.e. the red lines in fig. 21). We mentioned in sec 5.3 that this peak, for N=20, is roughly N/2. As the referee pointed out in his question point 2, this contrasts with the expected value of the total number of strings as derived in e.g. [1], which goes as \sqrt{N}.

However, there is actually no tension: our result of "N/2" is strictly a "small N" phenomenon, and cannot be extrapolated to larger N. Indeed, a careful calculation of the expected value (using e.g. the canonical ensemble given in (3.2)-(3.5) of [34]) confirms that for N=20, the expected value of the total number of strings is approximately 12.36. The same calculation at asymptotically large N, of course, gives the well-known behavior as \sqrt{N}.

This answers the referee's question in point 2. We have also corrected the statement about the typical state at large N that the referee refers to in his point 3.
(Note that in all other places that (the results of) model 3 are discussed, including the other (last) paragraph of sec 5.3, no analogous misleading statements are made.)​

---

## Round 3 · List of Changes

minor clarifications, updated the discussion on the D1-D5 system (in accordance with referee report)

You are currently on this page

Resubmission 1812.09328v3 on 28 March 2020

---

## Editorial Decision

published